


# Changes in the domestic heating fuel in Greece: effects on atmospheric chemistry and radiation

Eleni Athanasopoulou[1], Orestis Speyer[1], Dominik Brunner[2], Heike Vogel[3], Bernhard Vogel[3], Nikolaos Mihalopoulos[1,4], Evangelos Gerasopoulos[1]

[1]Institute of Environmental Research and Sustainable Development, National Observatory of Athens (IERSD/NOA), GR 152 36, Athens, Greece
[2]Laboratory for Air Pollution/Environmental Technology (EMPA), CH-8600 Dübendorf, Switzerland
[3]Karlsruhe Institute of Technology (KIT), 76344 Eggenstein-Leopoldshafen, Germany
[4]Environmental Chemistry Processes Laboratory, Department of Chemistry, University of Crete, Heraklion, P.O. Box 2208, 71003, Greece

*Correspondence to*: Eleni Athanasopoulou (eathana@noa.gr)

**Abstract.** For the past 8 years, Greece has been experiencing a major financial crisis which, among other side effects, led to a shift in the fuel used for residential heating from fossil fuel towards bio-fuels, primarily wood. This study simulates the fate of the residential wood burning aerosol plume (RWB smog) and implications on atmospheric chemistry and radiation, with the support of detailed aerosol characterization from measurements during the winter 2013-2014 in Athens. The applied model system (TNO-MACC_II emissions / COSMO-ART model) and configuration used, accurately predicts the frequent nighttime aerosol spikes (hourly $PM_{10}$ > 75 µg m$^{-3}$) and their chemical profile (carbonaceous components and ratios). Updated temporal and chemical RWB emission profiles, derived from measurements, were used, while the level of model performance was tested for different heating demand conditions, resulting to better agreement with measurements for $T_{min}$ < 9 °C. Half of the aerosol mass over the Athens basin is organic in the submicron range, 80% of which corresponds to RWB (average values during the smog period). Although organic particles are important light scatterers, the direct radiative cooling of the aerosol plume during the wintertime is found low (monthly average forcing of –0.4 W m$^{-2}$ at the surface). This is attributed to the timing of the smog plume, both directly –important interactions with the long-wave radiation during the nighttime emission- and indirectly, i.e. the mild effect of the residual plume on solar radiation during next day, due to the removal and dispersion processes.

## 1 Introduction

Biomass has been traditionally used as a fuel for residential heating in several regions with cold climates, such as the Alpine mountain range and Scandinavia (Yttri et al., 2005; Molnar et al., 2005; Puxbaum et al., 2007; Herich et al., 2014). The European incentives to use biomass fuels, under the assumption of biomass carbon neutrality, have caused a further increase of wood burning for heating purposes in several European cities (Bari et al., 2010; Gu et al., 2013; Waked et al., 2014;



Hovorka et al., 2015). Along with this trend, the financial crisis in Europe during the past nine years has about a significant increase in the market price of conventional fuels, primarily due to exorbitant taxation. In Greece, the respective increase on the non-solid fuel price was in the range of 40–60%. The high market price together with the limited financial capacity of many households has resulted in an excessive use of biomass for domestic heating (Sarigiannis et al., 2014, Fourtziou et al., 2017).

Several studies have focused on the impacts of residential wood burning (RWB) emissions on air quality. Fuller et al. (2013) reported on aerosol ($PM_{10}$) contributions from residential wood burning (hereafter also called 'RWB smog') in three major European cities: London, Paris and Berlin and found that, during winter, RWB makes an important contribution to breaches of the daily mean EU limit and $PM_{10}$ contribution may surpass that of road traffic. In Portugal, wood combustion is estimated to comprise 60% of residential energy use, but to account for almost 99% of domestic $PM_{10}$ emissions (Borrego et al., 2010). An emission inventory constructed for Greece by Fameli and Assimakopoulos (2016) showed that regarding residential emissions, 67% of the emitted CO in Greece originates from fireplaces. At the European level (EU-28), fuel combustion in the residential combustion sector (commercial, institutional and household) is the major source of primary $PM_{2.5}$ and $PM_{10}$, as well as black carbon (BC) emissions contributing 56, 40 and 46 % respectively (EEA, 2016). Residential combustion (mainly wood) accounts for 3-20% (2-30%) of ambient annual mean $PM_{10}$ ($PM_{2.5}$) levels in different European regions, with maximum contributions of up to almost 30% (30%) for the winter period mean value (Viana et al, 2016).

The type of areas examined regarding wood burning impacts on air quality, may range from a specific residential area or a suburb (Hellen et. al., 2008 and Yli-Tuomi et. al., 2015), to a large city such as London (Fuller et. al., 2014, Young D.E. et. al., 2015) and to a regional level, such as Flanders (Maenhaut et. al., 2012). They all agree on a substantial contribution of RWB to PM mass, even when the use is for secondary heating. Wood combustion in open fireplaces emits a mixture of soot (BC or EC) and organic carbon (OC), with the latter being the major component of RWB aerosol (Bolling et al., 2009). The contribution of wood combustion to OC is found up to 41% of OC in the $PM_{10}$ fraction during the wintertime in Zurich (Szidat et al., 2007) and up to 50% in Lombardy (Piazzalunga et al., 2011). In Switzerland, during winter, BC from RWB contributes on average 24–33% to measured BC levels, a noticeable high fraction as the contribution of wood burning to the total final energy consumption is less than 4 % (Herich et al., 2014). Saffari et al. (2013) showed the connection between wood burning and increased levels of PM during the winter in Thessaloniki, while Sarigiannis et al. (2014) further quantified an average increase in $PM_{10}$ (and $PM_{2.5}$) due to biomass burning of up to 43µg m$^{-3}$. In Athens, the intense use of open fireplaces and woodstoves for domestic heating during the last years has caused notable increases in the ambient PM concentrations, with the contribution from biomass combustion reaching up to 70% of the PM mass concentration during extreme pollution episodes. The relevant five-year measurements study showed also an increase of 80% in BC due to increased wood burning (Paraskevopoulou et. al., 2014). The chemical footprint of these smog events has been well established via specialized aerosol measurements, e.g. significant correlations between $PM_{2.5}$ and levoglucosan, or high fine mode particulate potassium, demonstrating that, during wintertime, wood burning in Athens could be responsible for $PM_{2.5}$ levels higher than 45 µg m$^{-3}$ (Fourtziou et al., 2017; Paraskevopoulou et al., 2014).



Modeling applications devoted to RWB are less numerous and primarily focused on source apportionment and sensitivity to emissions. Simpson et. al. (2007) pointed out that the missing wood-burning contributions may explain the discrepancies between simulations and observations for wintertime OC. Fountoukis et. al. (2014) estimated a decrease in fine organic aerosol (up to 60% in urban and suburban areas during winter), elemental carbon (30-50% in large parts of Europe) and

$PM_{2.5}$ mass (15-40% during winter in continental Europe), by replacing current residential wood combustion technologies with pellet stoves, underlining the high sensitivity to emissions. This increasing importance of RWB is also recognized by the revised higher estimates of the latest version of the TNO emission database (van der Gon et al., 2015), as far as particulate matter is concerned. The inventory indicated that about half of the total $PM_{2.5}$ emissions in Europe are carbonaceous aerosol and identified RWB as the largest organic aerosol source. Moreover, the authors note that while

emissions of particulate matter or carbonaceous aerosols are notoriously uncertain, a revised bottom-up inventory was constructed with emphasis on residential wood combustion, which was previously significantly underestimated.

There can be a multitude of impacts of the increasing carbonaceous aerosol contribution to PM as a result of increasing importance of RWB. Exposure to ambient particulate matter has been associated with a range of negative health effects, including increased morbidity and mortality from pulmonary and cardiovascular diseases (Bolling et al., 2009). Recent

epidemiologic studies have displayed the risks of exposure to increased levels of carbonaceous aerosols, a major constituent of wood smoke, revealing notable associations with the aforementioned diseases (Ostro et al., 2009; Lipsett et al., 2011; Krall et al., 2013). In a systematic review from short-term epidemiological and cohort studies, it was found that BC is a better indicator of harmful particulate substances from combustion sources than undifferentiated particulate matter mass and it may operate as a universal carrier of a wide variety of chemicals of varying toxicity (Janssen et al., 2012).

Another important aspect of wood burning particles and other aerosols is their direct and indirect effect on atmospheric physics and dynamics. Carbonaceous species, composed of both light-absorbing BC and light-scattering OC, are well recognized contributors to radiative forcing (RF) (Novakov et al., 2005), and the fraction of OC/EC (or OC/BC) becomes decisive with respect to the sign of contribution (Baumer et al., 2007). BC has been considered as the second most important climate forcing agent from human sources in the present-day atmosphere, behind carbon dioxide (Bond et al., 2013). Several

studies concentrated on the direct radiative effect of aerosols and their impact on ground temperature (Vogel et al., 2009; Stanelle et al., 2010; Bangert et al., 2012; Lungren et al., 2012; Athanasopoulou et al., 2013). Severe winter haze events over the heavily polluted North China Plain and over Eastern China were found to have a significant negative RF ranging from 20 to $140 Wm^{-2}$ (Gao et al., 2015; Zhang et al., 2015). Focusing on a carbonaceous-rich aerosol loading induced by wildfires in Greece, Athanasopoulou et al. (2014) revealed a negative impact on the surface radiative budget in the order of $10 \ Wm^{-2}$

(three day average) and a reduction of surface temperature by 0.5 K over land. The radiative effect of aerosols on longwave radiation is less examined. Stanelle et. al. (2010) concentrated on dust episodes over West Africa and found an average increase of $70 \ Wm^{-2}$ for the longwave radiative effect which was, however, nonlinear. Panicker et al. (2008) found a positive surface forcing of $6-9 \ Wm^{-2}$ over an urban environment in Prune, India. Finally, aerosols have an indirect effect on meteorology and climate by acting as cloud condensation nuclei (CCN) and ice nuclei (IN) in aerosol-cloud interactions





(Kanakidou et al., 2005, Bangert et al., 2011). To the best of our knowledge, there has been no study so far addressing the direct radiative effects of RWB smog.

The current study focuses on the winter 2013-2014 in Athens, when several severe RWB smog events were recorded. The comprehensive, online-coupled, modelling system COSMO-ART was used to quantify the effect of wood burning on aerosol

levels and aerosol chemistry, as well as on radiation during this period. Detailed chemical measurements of concentrations, speciation and fractions related strictly to wood burning, were available from the longest winter measuring campaign of RWB smog to date (e.g. Paraskevopoulou et al., 2014; Fourtziou et al., 2017). Within this paper we want to address the following topics: 1. Assessment of the emission data from RWB, 2. Contribution of RWB to the total aerosol load and chemistry, 3. Calculation of the optical properties of RWB aerosol and 4. Impact of RWB smog on radiation.

**2 Data and Methods**

**2.1 Experimental data**

Since winter 2012, several intensive campaigns have been designed and carried out at the official aerosol monitoring site of the National Observatory of Athens (NOA), at its central premises at Thissio (37.97 °N, 23.72 °E), in Athens' city center. Thissio is an urban background site, representative of Athens' average atmospheric load. Details about the site are provided

in Paraskevopoulou et al. (2015) and Fourtziou et al., (2017). For the needs of this modeling study, measurements were taken from the campaign in winter 2013-2014 (16 December, 2013 to 21 February, 2014), during which unique datasets of wood burning indicators were acquired (Fourtziou et al., 2017).

In particular, a time series of hourly $PM_{10}$ is used here taken from a $PM_{10}$ beta-attenuation analyzer. $PM_1$ chemical composition of non-refractory aerosol particles (organics, sulfate, nitrate, ammonium and chloride) was obtained by an

Aerosol Chemical Speciation Monitor – ACSM. $PM_1OC$ was obtained via thermal optical transmission technique, using a Sunset Laboratory Inc. (Oregon) carbon analyzer (see Paraskevopoulou et al., 2014) and $PM_1BC$ by a portable aethalometer (AE-42; Magee Scientific, at 7 wavelengths: 370, 470, 520, 590, 660, 880 and 950 nm). Details about the aforementioned measurements and techniques for the specific campaign can be found in Fourtziou et al., (2017). Among the results described there (and references therein) is the decomposition of the BC time series into BC associated with fossil fuel ($BC_{ff}$)

and wood burning ($BC_{wb}$), enabled by the wavelength dependence of the BC measurements. Standard meteorological parameters (relative humidity, temperature, rain, wind speed and direction) are measured on a routine basis at the station.

**2.2 Model framework and setup**

COSMO-ART is a regional atmospheric model which couples online meteorology, chemistry, and aerosol dynamics. COSMO is the operational numerical weather prediction model of the German and other European weather services (Baldauf

et al., 2011) and is used as a regional climate model in a modified version CCLM (Rockel et al., 2008). ART (Aerosols and Reactive Trace gases) is the chemistry extension of COSMO. Detailed descriptions of the model, the physic-chemical

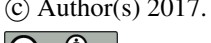



characteristics of the aerosol modes and the parameterizations of feedbacks of aerosols on radiation, temperature, cloud and ice condensation nuclei (CCN and IN), are given in Vogel et al. (2009), Bangert et al. (2011; 2012), and Rieger et al. (2014). The model domain chosen in this study is the extended area of Greece, centered on Athens, which is the region of interest. The setup of the current model application is largely based on the pilot study of the COSMO-ART application over the same

domain (Athanasopoulou et al., 2014) and is given in Table 1. The atmospheric pressure and precipitation parameters were optimized with respect to the high spatial resolution of the current application (0.025˚). The simulations were carried out for the period 17 December 2013 till 22 January 2014, providing hourly outputs directly comparable with the aforementioned in-situ, surface level measurements (Section 2.1). The first two days of all model runs (Table 2) were used as spin-up to dampen the effect of the constant initial conditions (e.g. for $SO_2$ and aerosol species).

Hourly anthropogenic emissions of gases (CO, $NH_3$, NMVOC, $NO_x$ and $SO_2$) and aerosols ($PM_{10}$, $PM_{2.5}$) are based on the TNO-MACC_II emission inventory for Europe (van der Gon et al., 2010; Kuenen et al., 2011; 2014; Denier van der Gon et al., 2011). Data are provided at a high-resolution ($7 \times 7$ km$^2$) and reflect emissions from industrial sources, road (and off-road), rail, air and other transport, waste treatment, agriculture and residential combustion.

The processing of this data set for COSMO-ART applications includes a source-specific speciation of NMVOCs and PM

(BC, OA, $SO_4$, rest), the application of time profiles (for diurnal, day-of-week and seasonal variability), and a mapping onto the simulation grid (Knote et al., 2011). For the present study, emissions representative of 2009 (latest year available in the inventory) were used with no further scaling to the simulation period. In the frame of the current study, the prescribed wood combustion emissions and aerosol optical properties (Vogel et al., 2009) have been revised, according to observational findings representative for the area and period of interest (Sect. 2.3 and 2.4).

**2.3 Modifications of the aerosol emissions from residential combustion**

Non-industrial (residential and agricultural) combustion is spatially distributed according to population density and the proximity to wood (Kuenen et al., 2014). For Greece, it is calculated that almost all mass (i.e. 98%) of the total $PM_{10}$ emissions from this source category reflects wood combustion (i.e. open fireplaces).

Unlike the emission data for the gaseous species, which are officially submitted by Greece (EEA, 2011), $PM_{10}$ (and $PM_{2.5}$)

emissions are not based on real data, but solely on model approximations (IIASA, 2012), thus significant improvements are feasible upon real data availability. For instance, changes in the temporal emission profiles can have a great impact on the air quality model results for Europe, especially with respect to residential combustion (Denier van der Gon et al., 2011). More specifically, the diurnal profiles suggested therein for the residential combustion sector, have a peak in the morning (black line in Figure 1a), while the common practice in Greece is to use fireplaces mainly in the evening. Furthermore, space

heating is more intense on weekends, opposite to the respective suggested weekly variation (black line in Figure 1b). Therefore, both the hourly and weekly cycles applied to wood combustion emissions were revised to better match common practices in Greece. In particular, assuming that the temporal variation of $BC_{wb}$ closely follows the temporal profile of the corresponding emissions during winter when the planetary boundary layer (PBL) dynamics is moderate, hourly observations



of this well-defined RWB index, normalized with respect to the average value, were used for the revision (green line in Fig. 1a). The average profile was derived from the long term monitoring of BC, during the 2013–2015 winter periods. In this way, the two primary peaks in the standard profile were replaced by a much more realistic profile, with an intense peak in the evening hours. In order to revise the original weekly cycle for residential wood combustion, the same data were used, but

only during the peak activity period of the fireplaces (20.00-02.00 LST). A representative day-of-week cycle was derived from the observations, based on the median values (to minimize the influence of possible outliers e.g. holiday weekdays). Again, the weekly cycle (green line in Figure 1b) is now consistent with the increased residential wood combustion in Athens during the weekends. It should be noted that the revised temporal cycles were applied only to the aerosol fraction of wood combustion (98% of total combustion aerosol emissions), while gases were assumed to be emitted in equal amounts by

wood (revised profiles) and non-wood fuel combustion (original profiles). Thus, the final profiles were calculated from the respective weighted means.

Furthermore, the original chemical profile of aerosol emissions from non-industrial combustion (20% BC, 40% OA and 40% others) was modified based on the chemical composition of aerosol spikes ($> 75 \mu g \ m^{-3}$, i.e. EU alert value for the whole population) during night time. Measurements show that the aerosol mass originating from fireplaces is primarily composed

of OA (80%), while the rest is equally partitioned into BC and other species. This doubling of the OA mass is in line with a recent study on particulate emissions, based on the latest rise in residential wood combustion in Europe (Denier van der Gon et al., 2015) and is related to the semi-volatile organic matter that is instantaneously formed after emissions are exposed to the cooler ambient temperature. Furthermore, the currently applied emission ratio of BC/OC (0.12), derived from the measurements at Thissio, is close to the average of 11 studies (0.16 ± 0.05) on emission factors for RWB (fireplaces),

presented in Szidat et al. (2007).

A representative map of the modified emission rates for the area of interest is shown in Figure 2. Combined with the temporal profiles, the hourly emission rates from wood burning for any hour and day of a typical winter week in Athens as represented in the revised TNO-MACC_II emissions database. As expected, emission values have a maximum at the urban core, where aerosol emissions from RWB represent around 90% of the total value. In particular, the hourly emission rate of

25 aerosol during a smog episode in the Athens city center is in the range of $3.5 – 4.5 \ kg \ hr^{-1} \ km^{-2}$.

The revised TNO database for Athens was used for the revised run of the current study (case 2 in Table 2). A comparative run using the original TNO-MACC_II database was also performed (case 1 or baseline) to evaluate the level of improvements achieved. It should be noted that an artificial increase in the mass of residential aerosol emissions due to the switch from residential heating from fossil fuels to wood burning was not attempted, as the comparison among model

outputs and observations during the smog period (Sect. 3.1.1.) confirmed the magnitude of the original daily $PM_{10}$ emission rates for residential combustion.



## 2.4 The aerosol optical properties

The properties of atmospheric particles (chemical composition, liquid water content, optical properties) determine their effects on short- and long-wave radiation (scattering and absorption). COSMO-ART incorporates prescribed values for the single scattering albedo, extinction coefficient (in $m^2\ g^{-1}$) and asymmetry factor for the five ultrafine aerosol mixtures it handles (pure soot, aged soot coated with soluble material in the nucleation and in the accumulation mode and purely soluble mixture in the nucleation and in the accumulation mode).

For the purpose of the present study, the calculation of the aerosol optical properties is based on observational data collected in Greece during the 2013-2014 RWB smog episodes. In particular, the average surface chemical composition of ultrafine aerosols in Athens (pure soot: 2.8 $\mu g\ m^{-3}$, water soluble mixture of sulfate, nitrate, ammonium and organics: 22.2 $\mu g\ m^{-3}$), the local relative humidity (50-70 %) and an average mixing layer height (600 m a.g.l., Gerasopoulos et al., 2017), were used to feed the OPAC software (Hess et al., 1998), which then - by applying the Mie theory - provides the respective optical properties for 61 wavelengths between 0.25 and 40 μm (grey lines in Figure 3). These values have been applied to the eight wavelength bands simulated by COSMO, which cover the spectral range from 0.25 to 104.5μm (green lines in Figure 3) and were used for the case 2.

These values differ from the ones used in Vogel et al. (2009; black and red lines in Figure 3). On the contrary, the smooth reduction with increasing wavelength in the single scattering albedo (SSA) of pure soot, as well as the abrupt decrease from 1 to 3 μm in the water soluble aerosol modes, are consistent with the findings of Mishra et al. (2014; 2015) for the polluted air masses over the Mediterranean, as well as by Takemura et al. (2002) for carbonaceous and sulfate aerosol.

## 3 Results

The period studied can be characterized as a relatively mild winter period. December 2013 was only the 34[th] coldest December of the period 1897-2014 (NOA records for Athens), with a daily mean temperature of 10.7°C and minimum temperatures ($T_{min}$) from 4.9 to 11.5 °C. January 2014 was the warmest January since 1897. The mean daily temperature was 12.5°C. Daytime peaks reached values above 16°C, while nighttime temperatures were greater than 10°C for 75% of the days. The coldest period of this month was 3-5 January, with $T_{min}$ from 7.3 to 8.2 °C. The relative humidity during the period was generally above 50% reaching values above 90% during several precipitation events, which took place on 26-27 December 2013, 31 December 2013 – 2 January 2014, 05-06 and 15-16 January 2014 (Figure 4a).

The wind speeds during this period were rather low (usually from the northern sector), preventing the dispersion of pollution. The average wind speed was 2.7 ms-1, with stronger winds (5 - 8 $ms^{-1}$) from the north direction on 17, 27, 31 December 2013 and 07, 15 and 19-22 January, 2014. An exception with strong (up to 8 ms-1) southeastern winds occurred on 21 January, when a Saharan dust intrusion took place. This event is well documented by satellite retrievals of Aerosol Optical Depth (https://giovanni.sci.gsfc.nasa.gov/giovanni/), as well as BSC-DREAM8b model outputs (http://www.bsc.es/ESS/nmmb_bsc-dust).





The meteorological conditions favored the accumulation of smog over Athens during the period 19 December 2013 – 05 January 2014 (mean maximum nighttime $PM_{10}$ measured concentrations of 103.7 µg m$^{-3}$). This time frame, excluding the rainy days mentioned above, constitutes the intense smog period of the current modeling study, analyzed in Sect. 3.1.1. An interesting case to study aerosol spatial fields (Sect. 3.1.2) is 4-5 January (from noon to noon), as this was the most polluted

smog event of the whole period (the maximum nighttime $PM_{10}$ hourly measured value was 167.7 µg m$^{-3}$) and showed a good model performance (the maximum modeled nighttime $PM_{10}$ value was 202.2 µg m$^{-3}$). A mild winter day (January, 7 noon – 8 noon, $T_{min}$= 10.6 °C) with low smog influence ($BC_{wb}$ equals 44% of $BC_{tot}$) according to measurements and predictions (Figure 4c), is also examined for comparison purposes, using results from the case 3.

The mean radiative aerosol effect under smog conditions is discussed in Sect. 3.2, together with the maximum impact during

the intense smog event defined above. Cloudless conditions during this day with no interferences by cloud-radiation interactions as well as the good representation of aerosol observations gives credibility to the model results with respect to the direct radiative effects of wood burning smog.

### 3.1 Impacts of residential wood burning (RWB) on atmospheric aerosol mass and chemistry

### 3.1.1 Aerosol model performance under smog influence

The applied model system as well as its configuration and methodology for the smog period simulation is here examined and evaluated through: (a) the reproduction of the mean and peak values of mass concentrations of $PM_{10}$, $PM_1OA$ and $PM_1BC$ measurements, (b) the degree of correlation with the respective observations, (c) the values of representative aerosol ratios (RWB fraction, OC/BC, BC/TC, TC/$PM_{10}$, OA/$PM_{10}$ and BC/$PM_{10}$) and (d) the reproduction of the observed diurnal cycles of $PM_{10}$, $PM_1OA$ and $PM_1BC$. The model's ability to reproduce the mean values of mass concentrations is quantified by

calculating the Mean Bias (MB), the correlation coefficient (r2) and the Mean Absolute Normalized Gross Error (MANGE), for cases 1 and 2 (baseline and revised simulation). For the peak values, the Paired Peak Estimation Accuracy (PPEA) is used. A summary of the model skills is given in Table 3, while the mathematical formulation of the applied statistical parameters is given in Appendix A.

The mean value of $PM_{10}$ observed during the intense smog period was 45.2 µg m$^{-3}$, which is nicely captured by case 2 (50.8

25  µg m$^{-3}$), but overestimated by the baseline (58.7 µg m$^{-3}$). Although the mean bias between measurements and the two model runs is similar, the correlation of hourly data with observations is greatly improved in case 2 with an r$^2$ of 0.66 compared to only 0.24 in case 1. This is related to the revisited temporal cycles of emissions, which represent the use of wood for space heating purposes in Athens much more realistically. In particular, the observed nighttime $PM_{10}$ peaks were on average 103.6 µg m$^{-3}$, more than two times higher than the mean daytime maximum (47.8 µg m$^{-3}$). Case 1 failed to represent these values

(e.g. morning PPEA equals +173 %), due to the pre-crisis space heating (mainly office) during the working hours (cf. black line in Fig. 1a). In contrast, case 2 is greatly improved (80% and 23%, respectively for morning and nighttime PPEA), showing realistic diurnal profiles during the winter period (under crisis), when the mean hourly concentrations of modeled





and measured $PM_{10}$ ranged from 20 (morning) to 100 μg m$^{-3}$ (night; Figure 5a). Overall, the revised run improved more than the 70% of the day and nighttime $PM_{10}$ peaks during the intense smog period.

As evident in Table 3, the most important effect of the incorporation of emission revisions into the model runs is the improved representation of the mean submicron mass fractions ($PM_1$) of organic (OA) and black (or elemental) carbon (BC).
In particular, r$^2$ equals 0.73 and 0.53 for the two components, respectively, both statistically significant at the 99% c.l. The chemical composition of oil combustion (central heating) greatly differs, thus the baseline run (chemical profiles suggested by the TNO) fails to represent winters during crisis years. Similarly to $PM_{10}$, the hourly peaks of both submicron species cannot be reproduced by the baseline run. In case 2, the mean nighttime peak value of $PM_1OA$ is predicted 69.1 μg m$^{-3}$, close to the measured value of 85.1 μg m$^{-3}$, while the outputs from case 1 are unrealistically low (19.6 μg m$^{-3}$). The PPEA values for both carbonaceous species are significantly lower for case 2, which leads to the improvement of the half $PM_1OA$ and of all $PM_1BC$ the daytime peaks during the intense smog period.

The mean diurnal cycles of both carbonaceous species (Figure 5b and c) are satisfactorily represented by case 2, while case 1 is strongly biased, especially during daytime for $PM_1BC$ (about 5-fold overestimation) and nighttime for $PM_1OA$ (about 3-fold underestimation). The mean hourly evening peak of $PM_1OA$ in the observations was 74.4 μg m$^{-3}$ at 23:00 UTC and nicely captured by the revised run (62.7 μg m$^{-3}$ at 22:00 UTC). The respective findings for $PM_1BC$ are 9.3 μg m$^{-3}$ at 21:00 UTC in the observations and 11.3 μg m$^{-3}$ at 22:00 UTC in the simulation. The largest diurnal amplitude during a typical smog day was observed for organics for which the mean minimum (daytime) concentration was around 10 times lower than the mean maximum (nighttime), while the day-night difference for $PM_1BC$ was only by a factor of 7. This is related to the role of wood burning in each mass fraction, which is discussed below.

The fact that the model biases in the RWB fraction of the BC (case 2 – case 3), OC/BC, BC/TC, TC/$PM_{10}$, OA/$PM_{10}$ and BC/$PM_{10}$ are only small, indicates that the aerosol composition was accurately represented in case 2 during the intense smog period. The mean values from predictions and observations at the station Thissio are presented in Table 3. All model findings are in very good agreement with the observations in the city center. In particular, the mean fraction of RWB particles in the total mass of the predicted $PM_1BC$ was 42 (45) %. This is a daily mean, i.e. the hourly fraction ranged from 25% (daytime) to 70% (nighttime) both in the measurements and the predictions (Figure 5c). RWB smog was estimated to comprise up to 50% of $PM_{10}$ concentrations during the intense smog period in Athens and to account for almost 80% of $PM_1$ organics (nighttime peak up to 90%; Figure 5b). It is interesting to note that in the case of organics, wood combustion continues to outweigh all other sources during the whole day (daytime value of more than 60%). Thus, the aerosol chemical composition during the economic crisis is completely altered with respect to the chemical profile of wintertime aerosols beforehand.

The OC/BC ratio calculated from the baseline run (1.1) is rather unrealistic, because it reflects typical urban environments, i.e. intense traffic, low biomass burning and limited regional contribution of aged aerosol. In contrast, both measurements (2.9) and revised predictions (2.8) are highlighting the influence by biomass burning and high secondary formation rates of organics (Szidat et al., 2009; Pio et al., 2011; Gianini et al., 2013; Airuse, 2014). The carbonaceous aerosol dominates $PM_{10}$



(62 %) in wood burning conditions, more than 70% of which corresponds to organics. The latter makes up half of the total $PM_{10}$ mass. Again, measurements and revised predictions are in agreement. The comparison with available measurements during increased wood burning in the Alpine area (Szidat et al., 2009) reveals similarities for the ratios EC/TC and EC/$PM_{10}$, while OC/BC and TC/$PM_{10}$ showed somewhat higher values during those experiments (cf. Table 3).

Overall, the exploitation of specialized, systematic measurements of $BC_{wb}$ has been crucial for updating the conventional TNO-MACC_II emission database and a necessary step to accurately reproduce the aerosol pollution in Athens during the financial crisis. The revised temporal cycles and chemical profiles of the emissions from residential combustion significantly improved aerosol predictions, especially during the peak hours (daytime for $PM_{10}$ and PM1BC and nighttime for $PM_1OA$). Outliers (unrealistically high model values) occurred mainly during weekends with high nighttime temperatures outside the
intense smog period (e.g. January, 12 and 18, cf. Figure 4) and are further discussed in Sect. 3.1.3.

### 3.1.2 Representative spatial aerosol fields

Figure 6 depicts the spatial distribution of the daily mean surface aerosol concentrations ($PM_{10}$, $PM_1OA$ and $PM_1BC$) and corresponding fractions (TC/$PM_{10}$, RWB and OC/EC), over the greater Athens area during the selected smog (left column) and mild (right column) event.

As seen, $PM_{10}$ levels during these two events differ by more than a factor two (contours in Figure 6a and b), with concentrations during the smog event reaching 140 µg m$^{-3}$ within the urban core, exceeding the EU alarm value for emission measures (100 µg m$^{-3}$), and being above the standard daily EU limit (50 µg m$^{-3}$) over the entire Athens basin. Concentrations during the mild day reach up to 60 µg m$^{-3}$, again exceeding the daily EU limit in the city center, which demonstrates the impact of RWB on air pollution throughout the whole winter period. This is related not only to the triggered accumulation of
pollution due to meteorology and topography of Athens, but also to the secondary organic aerosol formation due to RWB. Indeed, the largest fraction of $PM_{10}$ during the smog event is composed by carbonaceous matter (TC up to 80%, isolines in Figure 6a), 80% of which being organic. This is even more clear in Fig. 5c, where organics are elevated over the whole basin during the smog event (up to 75 µg m$^{-3}$) and the RWB fraction constitutes between 60% (city outskirts) and 80% (city center) of $PM_1OA$. During the mild period, in contrast, the air pollution is mainly composed by the rest of the aerosol species
(sulfates, ammonium, nitrates, rest; isolines in Figure 6b).

$PM_1BC$ reach very high levels under the RWB influence over the whole basin (daily values from 10 to 18 µg m$^{-3}$; Figure 6 e), the 60% of which corresponds to $BC_{wb}$ (not shown). On the contrary, $PM_1BC$ is found below 8 during the mild day (Figure 6f). The surface gradient of the OC/BC ratio is very smooth, i.e. the effect of RWB on organic production and formation, is regional and independent of the local peaks. This pattern seems to be stable in time, i.e. the high OC/BC value
(over 2.5) during the smog episode is similar to the mean value during the extended smog period (Table 3). Values around unity are found over the urban core, under typical heating and traffic-induced conditions (isolines in Figure 6f).

Overall, the aerosol levels at Thissio (all three RWB-affected components) can be considered representative of an extended urban area around the city center. The peaks observed at the urban core, as revealed from this model application, are





somewhat displaced from the exact location of the site. This finding is in line with the characterization provided by Gratsea et al. (2016), who used CO measurements from several sites (NAPM regulatory monitoring network of Athens), to characterize Thissio as an urban background site, not intensively affected by local traffic and representative of the average background pollution conditions in Athens.

In order to further support the representativeness of the site in a more quantitative way, we calculated the mean, minimum and maximum values of a greater urban area (118 km$^2$, 15 cells; SW corner: 37.94˚, 23.67˚) and compared them to the model results at the grid point of Thissio. Indeed, the temporal mean (19 December – 22 January 2014) point values of PM$_1$BC, PM$_1$OA and PM$_{10}$ (8.4, 39.0, 69.2 μg m$^{-3}$) are very close to the spatio-temporal averages over the extended urban domain (7.4, 37.8 and 64.9 μg m$^{-3}$, respectively; difference in the order of 3 - 7%). Furthermore, the hourly point values neither

exceed the area peaks nor fall below the minimum values, while they correspond to 66-68% of the maximum hourly concentrations found in the selected area. Lastly, the linear correlation among the site and the mean domain values for all aerosol components is high (r$^2$ above 0.73, N=817) and close to the 1:1 line (slopes from 1.03 to 1.11 and intercepts below 0.8) for all species.

### 3.1.3 Heating demand and model bias

As already pointed out in Sect. 3.1.1, aerosol concentrations were occasionally overestimated during mild winter weekends (concurrent green spikes and orange lines in Figure 4a). In order to examine whether this is valid for the whole mild period (5-22 January, 2014), the respective mean PM$_{10}$ levels (T$_{min}$> 8-9° C) were compared with the daily cycle predicted from case 3 (Figure 7). This run is indeed realistic, i.e. the actual wood burning during mild winter conditions in Greece is overestimated by the TNO-MACC_II database and by the subsequent atmospheric simulations (green line from 5 January

and onwards; Figure 4a). Thus, we attempted to estimate a relation between mass concentrations and model bias with temperature conditions. This was performed by introducing the heating energy demand (HD) term, namely the energy needed to heat a home located in Athens. It is defined as the degrees below the base temperature, which is here chosen as 15.5 °C (Carbon Trust, 2006).

Indeed, the observed hourly concentration levels of submicron aerosol in the city center of Athens during wintertime tend to

25 increase with decreasing temperature (black dots in Figure 8) and PM$_1$OA nighttime spikes are above 100 μg m$^{-3}$ only when air temperature falls below 8-9 °C. The explanation is two-fold: during cold weather conditions, firstly, the need for heating increases and thus all means of heating including fireplaces and woodstoves are expected to maximize, resulting in high RWB emissions. Secondly, the same conditions coincide with reduced vertical mixing over Athens with mixing layer heights varying between 200 and 400 m, which support the accumulation of air pollutants near the surface.

In order to identify the link between the model discrepancies and the actual space heating demand in Athens, the trend of the nighttime PM$_1$OA model bias and the daily maximum heating demand (at the T$_{min}$ of each day) was examined for the whole period excluding the rainy events (green dots in Figure 8). There is a significant correlation between the two (r$^2$ = 0.49), which shows that when the demand for heating decreases (i.e. at milder temperatures) the model tends to overestimate the



aerosol concentrations. In particular, all model bias from +25 μg m$^{-3}$ and above occur only when the HD is low (<6.5 °C), i.e. when nighttime air temperatures are above 9 °C. Inversely, during the cold days (T$_{min}$ below 8 °C) that the HD is increased (>7.5 °C), and consequently no model overestimation occurs.

This analysis shows the limitations of using average temporal profiles for the calculation of emissions from residential heating to feed model simulations. Evidently, the TNO-MACC_II residential wood combustion aerosol emission rates (using the updated temporal cycles proposed by the current study) are 'ideal' under typical winter conditions that lead to moderate-increased residential heating demand and during stay-at-home days (e.g. weekdays, holidays). It should be noted that this study has focused on the crisis period, i.e. a switch occurs in the domestic heating fuel from heating oil to wood, and that for a pre-crisis period or significant change in dominant heating fuels, the TNO-MACC_II emissions from residential heating should be further adjusted.

## 3.2 Impact of RWB smog on radiation

The effect of aerosols on total surface radiation during the simulated winter period was approximated by comparing case 2 with case 4. This is presented as a monthly mean in Figure 9a. As expected, the effect of total aerosol over the extended area of Athens is negative, i.e. the scattering of radiation by particles (organics, sulfates etc.) outweighs absorption by black carbon and the reflection of short-wave radiation (-1.9 Wm$^{-2}$) outweighs the enhanced trapping of long-wave radiation (+0.5 Wm$^{-2}$). Nevertheless, the mean monthly effect does not exceed -1.4 Wm$^{-2}$ at the urban core and by comparing the cases 2, 3 and 4 it is found that only 30% of this effect corresponds to particles from RWB. This means that the mean direct radiative effect (DRE) due to RWB at the surface does not exceed the value of -0.4 Wm$^{-2}$.

This small negative effect of particles on radiation is explained by the fact that most of the RWB emissions occur during nighttime when only the long-range terrestrial radiation is reflected (positive DRE values up to +0.4 Wm$^{-2}$). The role of timing on the aerosol radiative impacts has already been analyzed for dust storm over the eastern Mediterranean (Remy et al., 2015), which stresses the important positive feedback between aerosol and meteorology during nighttime. Besides, the residual plume during the next day is weakened, due to removal and dispersion processes. Indicatively, the nighttime PM$_{10}$ mean during the RWB smog episode (4-5 January, 2014) reached 145 μg m$^{-3}$, while the daytime mean was 98 μg m$^{-3}$, 50% of which corresponding to RWB particles. By removing the absorbing BC aerosols (14 μg m$^{-3}$), it is found that during the peak smog event of the studied period only 35 μg m$^{-3}$ corresponded to aerosols scattering in the short-wave, causing a DRE of -0.8 Wm$^{-2}$ at the urban core (Figure 9b).

Finally, we performed a run using the optical properties of aerosol from Vogel et al. (2009), which was compared to cases 2 and 3. The differences in the DRE were found to be negligible (not shown). This is explained by the fact that most of the differences that were imposed by the local conditions to the aerosol properties (cf. Sect. 2.4), correspond to the pure soot mode. Nevertheless, the atmospheric particles over an urban area correspond mainly to aged soot (coated with soluble material), a mode whose aerosol properties by Vogel et al. differ from the ones of this study on the long-wave part of the





spectrum (cf. Figure 3). However, the wavelength range above 2 µm is unimportantly affected by particles, as explained above. Thus, the short-wave scattering in both runs did not change significantly, leading to similar findings for DRE.

## 4 Conclusions

This study examines the impacts of increasing use of wood burning for domestic heating on air pollution through model-based analysis of a case study in Athens, Greece, as an immediate consequence of the ongoing economic crisis. A cold period ($T_{min}$ < 10 °C) with intense RWB smog conditions (nighttime $PM_{10}$ hourly peaks from 75 to 173 µg m$^{-3}$) was selected for the characterization of the aerosol levels and chemistry over the inner city center. Two representative events were distinguished for an analysis of the spatial representation of the smog plume during a cold ($T_{min}$ = 8.2 °C) and a mild ($T_{min}$ = 10.6 °C) winter day, with high and low nighttime peaks at the Thissio station in the city center ($PM_{10}$ at 173 and 59 µg m$^{-3}$, respectively).

The daily mean $PM_{10}$ concentration at Thissio during the RWB smog period derived from the measurements and the model (in parenthesis) was 45.2 (50.8) µg m$^{-3}$. The respective values for the $PM_1OA$ and BC were 28.4 (34.4) and 4.6 (7.1) µg m$^{-3}$. $PM_{10}$ levels during the smog (cold) event exceeded the EU daily limit for the whole Athens basin (model outputs). This finding is primarily related to RWB particles, which comprised 50% of the total $PM_{10}$ levels. The impact of RWB on air pollution persisted throughout winter, i.e. exceedances occurred also during the mild days, but confined to the urban core.

According to a spatio-temporal analysis of the current model results, the concentration values at Thissio are representative of an extended urban area, i.e. of the average pollution conditions in Athens. The day-to-night differences in the concentrations of different aerosol components were found to be very large (5-, 10- and 7-fold for $PM_{10}$, $PM_1OA$ and $PM_1BC$, respectively) because of the intense RWB activity during nighttime and the changes in PBL. In particular, the observed (simulated) peaks in $PM_{10}$, $PM_1OA$ and $PM_1BC$ between 21:00 and 23:00 UTC were on average 90.5 (100.2) µg m$^{-3}$, 74.4 (62.7) µg m$^{-3}$ and 9.3 (11.3) µg m$^{-3}$, respectively. These values correspond to almost 70% of the maximum hourly concentrations revealed by the model for the extended urban area, further supporting that measurements at Thissio represent urban background conditions over the Athens basin.

The carbonaceous component (OA+BC) of $PM_{10}$ in Athens during the smog period reached 62 (61) %, half of which corresponding to organic matter. This means that RWB completely alters the chemical profile of $PM_{10}$ over Athens during wintertime, as the RWB smog fraction of $PM_1OA$ (according to the simulation) reached from 60% during daytime to 90% during nighttime and $BC_{wb}$ accounted for 25% during daytime to 70% during nighttime of $BC_{tot}$ (according to both the measurements and simulation). The mean OC/BC ratio at Thissio from measurements (model outputs) was 2.9 (2.8), characteristic of the RWB influence and secondary formation of organics. Again, this value is representative for the whole Athens basin during wintertime, according to the spatio-temporal analysis of the model outputs.

Interestingly, the earth radiative budget was on average not altered significantly despite these high aerosol loads due to a compensation of long-wave and short-wave effects and the fact that RWB emissions are concentrated on nighttime hours





when only long-wave effects are present. Overall, the mean direct radiative effect of wintertime RWB smog was estimated as low as -0.4 Wm$^{-2}$.

For the model to properly capture the wintertime aerosol values observed in Athens, a revision of the residential combustion emission sector was necessary. This was facilitated via a unique, long-term dataset of the wood burning indicator (BC$_{wb}$) obtained at Thissio station. In particular, the TNO-MACC_II RWB emissions for particulate matter were revised with respect to their temporal and chemical profiles. The updated model configuration was found to greatly improve the prediction of smog episodes during wintertime. Thus, human health implications, as well as policy making, when the fireplaces are in use to cope with high heating demand conditions (HD > 7.5 °C), can be satisfactorily estimated and planned with the aid of such a tool. For mild winter conditions (T$_{min}$ > 8-9 °C), a post-processing of model results according to the linear regression between HD and model bias, can further improve the quality of the model system. Alternatively, an interactive treatment of RWB emissions, i.e. their online adjustment according to the actual temperature conditions of the simulation period, is proposed as a means to further enhance the reliability of operational forecasts of online-coupled atmospheric models.

## 5 Appendix A

The mean absolute normalized gross error (MANGE) is calculated by A1 and the correlation coefficient (r) by the A2:

$$MANGE = \frac{1}{N}\sum_{i=1}^{N}\frac{(E_i - O_i)}{O_i}100\%$$ (A1),

$$r = \left[\frac{\sum_{i=1}^{N}(E_i - \overline{E})(O_i - \overline{O})}{\sum_{i=1}^{N}(E_i - \overline{E})\sum_{i=1}^{N}(O_i - \overline{O})}\right]$$ (A2),

where $E$ is the estimated (modeled) and $O$ is the observed value of each parameter, paired in space and time for each $i$ of N data pairs. $\overline{E}$ and $\overline{O}$ are the mean values of estimations and observations, respectively.

The paired peak estimation accuracy (PPEA) is calculated as below:

$$PPEA = \frac{1}{N}\sum_{i=1}^{N}\frac{(EM_i - OM_i)}{OM_i}100\%$$ (A3),

where $EM$ is the estimated (modeled) and $OM$ is the observed peak one-hour value of each parameter, paired in space and time for each $i$ of N data pairs.

## 6 Acknowledgements

This work has been financially supported by the project ACTRIS-2, and the project THESPIA of the action KRIPIS (GSRT). Project ACTRIS-2 Integrating Activities (IA) has received funding from the European Union's Horizon 2020 research and



innovation programme (grant agreement No 654109). THESPIA was financed by Greece and the European Regional Development Fund of the EU in the frame of NSRF and the O.P. Competitiveness and Entrepreneurship and the Regional Operational Program of Attica. The computational time spent for this study is granted from the Greek Research & Technology Network (GRNET) in the National HPC facility - ARIS – under project ID ACRA. We appreciate the German Weather Service (DWD) for providing access to their forecast data records. We acknowledge the help from NOA colleagues responsible for the operation of the aerosol site at Thissio (Drs E. Liakakou, Dr. A. Bougiatioti, Dr, B. Psilogou, Dr. M Lianou, Msc I. Stavroulas and Msc L. Fourtziou),, our communication with Dr. H. Denier van Der Gon (TNO) with respect to emissions and Ulrich Schättler (DWD) for his continuous support on the configuration of the COSMO model.

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



**Table 1: Main characteristics of the COSMO-ART configuration and the current model setup. Italic fonts indicate emissions not considered in the current model runs.**

| Formulation / Mechanisms | COSMO-ART |
|---|---|
| Chemical mechanisms | Gaseous chemistry: RADMKA (Stockwell et al., 1990; Vogel et al., 2009) |
| | Inorganic aerosol chemistry: ISORROPIA II (Fountoukis and Nenes, 2007) |
| | Organic aerosol chemistry: VBS (Athanasopoulou et al., 2013) |
| | Wet scavenging and liquid-phase chemistry (Knote and Brunner, 2013) |
| Dynamic aerosol processes | Modal Aerosol Dynamics Model for Europe: MADEsoot (Riemer et al., 2003) |
| | (11 overlapping log-normally distributed modes) |
| Interaction between aerosols and meteorology | Aerosol effects on radiation and temperature: GRAALS (Ritter and Geleyn, 1992) |
| | Aerosol activation (CCN) and ice nucleation (IN) parameterizations (Bangert et al., 2011, 2012) |
| Initial and boundary conditions | Meteorology: COSMO model application over Europe (7×7 km), driven by the German Weather Service GME global model (Majewski et al., 2002) |
| | Gaseous species (CO, $HNO_3$, $NH_3$, $NO_x$, $NO_3$, $O_3$, NMVOC): global outputs (2.5°×1.9°) from the MOZART model (Emmons et al., 2010) |
| | Aerosol species: 1 µg m$^{-3}$ for sulfates, 0.1 µg m$^{-3}$ for nitrates, 0.37 µg m$^{-3}$ for ammonium and 1 µg m$^{-3}$ for secondary organic aerosol (SOA) in the surface layer[*] (Athanasopoulou et al., 2013) |
| Input data | Anthropogenic, agricultural emissions (prescribed): TNO-MACC_II (Denier van der Gon et al., 2010; Kuenen et al., 2011, 2014), |
| | *Fire emissions (prescribed): GFED v.3 (van der Werf et al., 2010),* |
| | Biogenic activity (isoprene, a-pinene, a-limonene), *Desert dust (3 aerosol modes)*, Sea-salt (3 aerosol modes) and DMS production, p*ollen and volcanic ash uplift and transport*: online calculation (Vogel el al., 1995, 2006; Lundgren et al., 2013; Nightingale et al., 2000; Zink et al., 2013; Vogel et al., 2014). |
| Vertical grid | 40 levels (from surface to ca. 23 km; first layer is ca. 20 m thick) |
| Horizontal domain | Greater area of Greece (0.025° × 0.025°) |



**Table 2: Description of modeling runs performed by the current COSMO- ART application from 17 December, 2013 to 22 January, 2014.**

| | Case | Description | Objective |
|---|---|---|---|
| 1 | Baseline | Original RWB emission data | Evaluation of the modified emission data from RWB in Greece |
| 2 | Revised[*] | Revised wintertime simulation | Aerosols over Athens during an intense RWB smog period |
| 3 | RWB-free[*] | Emissions from all residential combustion fuels, but wood | Contribution of RWB to the total aerosol load and chemistry and impact of RWB smog on radiation |
| 4 | Feedback-free[*] | The interaction between aerosols and meteorology is switched off | Impact of total aerosol on radiation |

[*]modifications in residential wood combustion emissions and in aerosol optical properties (cf. Sect. 2.3, 2.4)





**Table 3: Mean values and prediction skill metrics of the aerosol concentrations (and selected mass fractions) against ground measurements (at Thissio) during the RWB smog period (19 December, 2013 – 05 January, 2014) in Athens. Numbers in bold represent the calculated statistics. Italic letters indicate maximum values and statistics.**

| Component | Parameter | Observations | Case 1 (baseline) | Mean hourly bias | r² | MANGE (%) | Case 2 (revised) | Mean hourly bias | r² | MANGE (%) |
|---|---|---|---|---|---|---|---|---|---|---|
| **PM₁₀** µg m⁻³ (332 samples) | daily mean (smog period) | 45.2 | 58.7 | **15.0** | **0.24** | **95** | 50.8 | **19.3** | **0.66** | **83** |
| | std. deviation | 33.1 | 28.8 | | | *PPEA (%)* | 32.7 | | | *PPEA (%)* |
| | *mean daytime maximum* | *47.8* | *115.5* | | | *173* | *78.1* | | | *80* |
| | *mean nighttime maximum* | *103.6* | *80.9* | | | *-8* | *110.1* | | | *23* |
| % | RWB fraction | | | | | | 51 | | | |
| **PM₁ OA*** µg m⁻³ (333 samples) | daily mean (smog period) | 28.4 | 18.8 | **-8.2** | **0.39** | **65** | 34.4 | **6.6** | **0.73** | **118** |
| | std. deviation | 30.9 | 8.6 | | | *PPEA (%)* | 22.4 | | | *PPEA (%)* |
| | *mean daytime maximum* | *23.5* | *23.9* | | | *119* | *39.3* | | | *103* |
| | *mean nighttime maximum* | *85.1* | *19.6* | | | *-58* | *69.1* | | | *9* |
| % | RWB fraction | | | | | | 78 | | | |
| **PM₁ BC** µg m⁻³ (212 samples) | daily mean (smog period) | 4.6 | 9.9 | **5.2** | **0.47** | **236** | 7.1 | **2.5** | **0.53** | **121** |
| | std. deviation | 3.8 | 6.7 | | | *PPEA (%)* | 4.7 | | | *PPEA (%)* |
| | *mean daytime maximum* | *14* | *27* | | | *590* | *15* | | | *258* |
| | *mean nighttime maximum* | *5.2* | *12.6* | | | *63* | *11.3* | | | *38* |
| % | RWB fraction | 45 | | | | | 42 | | | |
| **OC/BC** | daily mean (smog period) | 2.9 | 1.1 | | | | 2.8 | | | |
| **BC/TC** | daily mean (smog period) | 28 | 47 | | | | 27 | | | |
| **TC/PM₁₀** | daily mean (smog period) | 62 | 48 | | | | 61 | | | |
| **OA/PM₁₀** | daily mean (smog period) | 50 | 32 | | | | 51 | | | |
| **BC/PM₁₀** | daily mean (smog period) | 11 | 17 | | | | 11 | | | |

*Organic aerosol (OA) predictions are divided by 1.6 (Turpin and Lim, 2001), to extract the carbon mass (OC) used for the calculation of the OC/BC ratio.



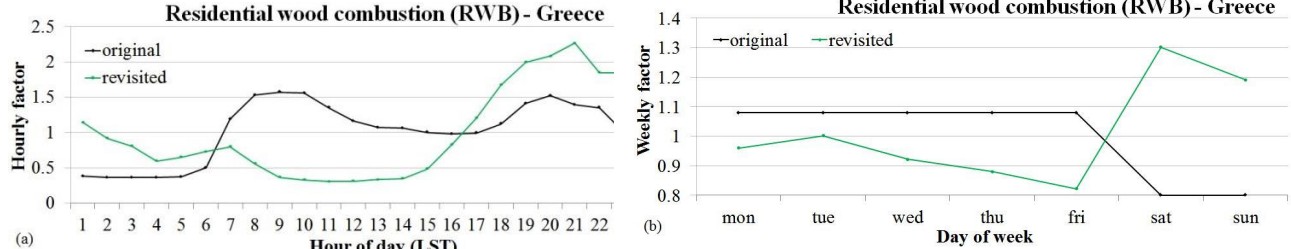

**Figure 1: Original (black line) and revisited (green line): (a) diurnal and (b) weekly profiles, applied to the emissions from residential wood combustion (RWB) in Greece (TNO-MACC_II database; Denier van der Gon et al. 2011).**





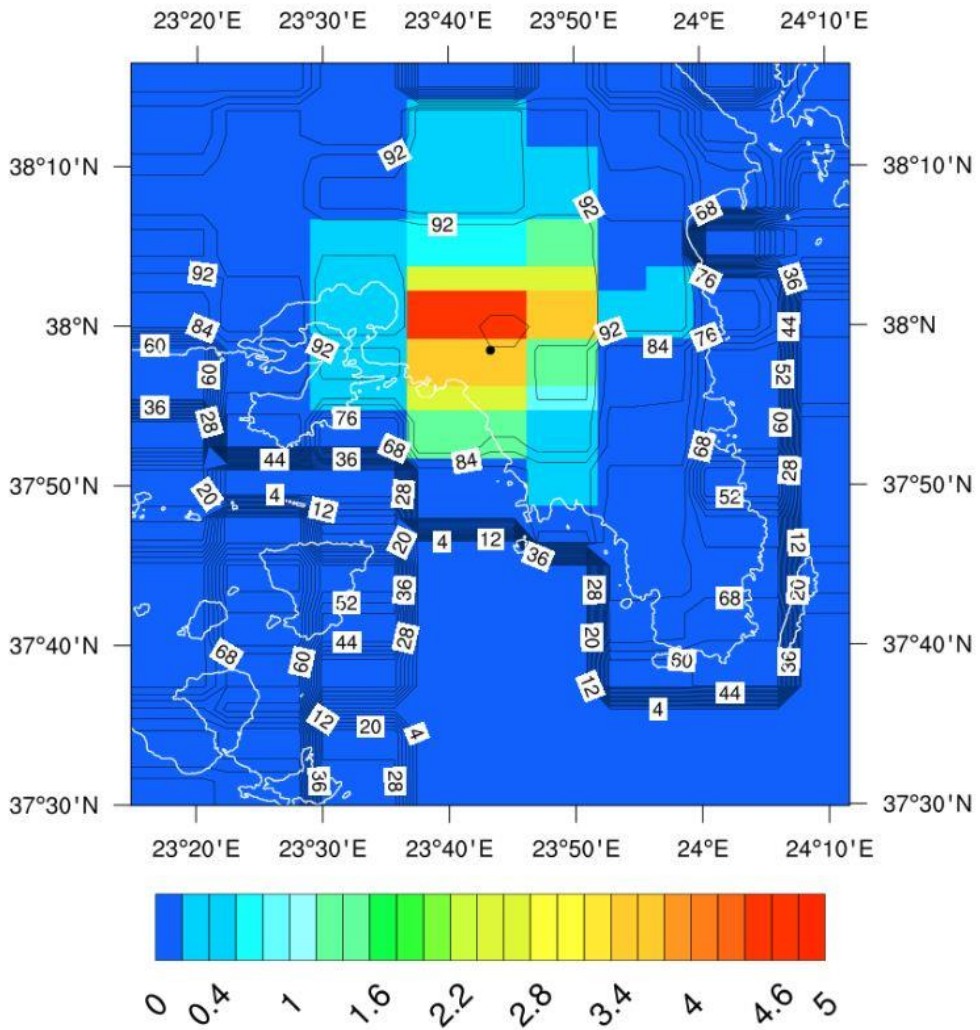

**Figure 2: RWB aerosol emission rates (kg hr$^{-1}$ km$^{-2}$) (color contours) and their fraction of the total aerosol emissions (isolines in %) for a night hour of a weekday (Tuesday, 21.00 UTC), as retrieved from the revised emission (TNO-MACC_II) data. It is noted that the spatial resolution of the emissions (0.125° × 0.0625°) is coarser than that of the model configuration (0.025° × 0.025°). The city center of Athens is shown with the black mark.**



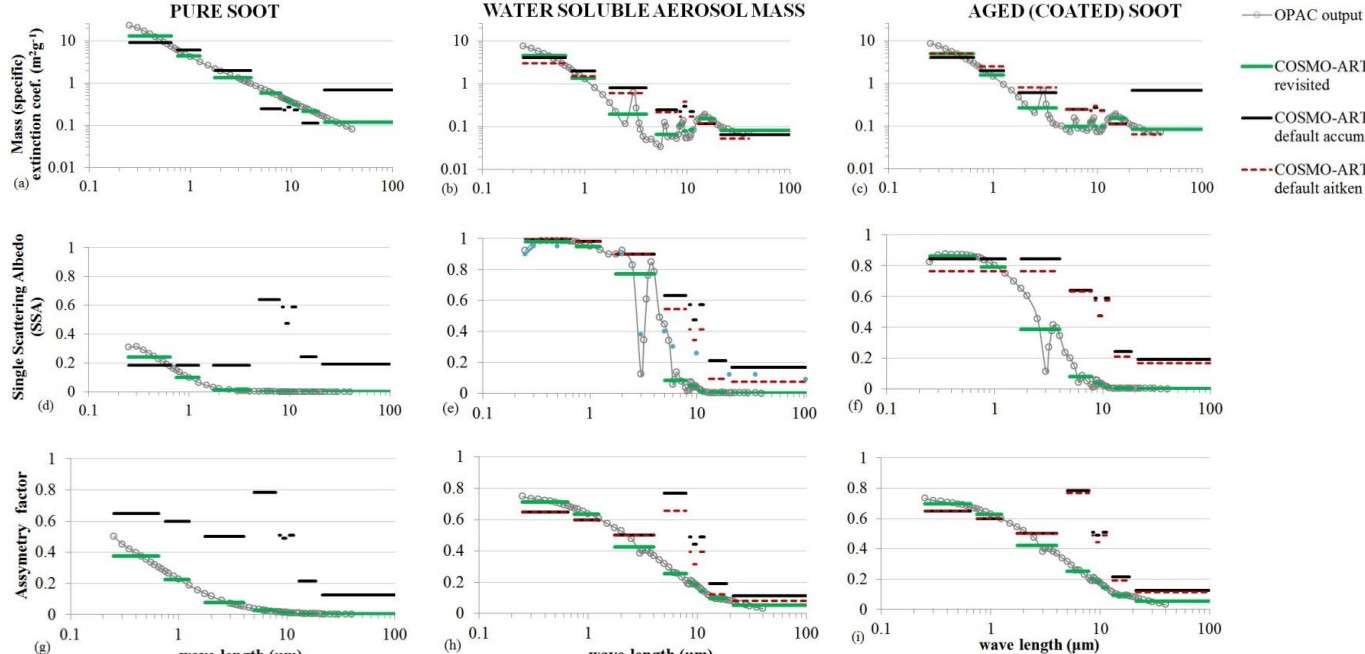

**Figure 3: Specific extinction coefficient (a – c), single scattering albedo (d - f) and asymmetry factor (g - i) per wavelength, for different mixtures of submicron aerosol (pure soot, water soluble mass and coated soot): from the COSMO-ART configuration described in Vogel et al. (2009; black and red lines for the accumulation and nucleation mode, respectively. Soot is represented by a single mode), from the OPAC algorithm (grey lines), and the OPAC values adjusted to the COSMO-ART bands (green lines).**





**Figure 4:** Time-series of the hourly concentrations (μg m⁻³; left axis) of the: (a) total PM₁₀ mass, (b) submicron organic aerosol (PM₁OA) and (c) submicron black carbon (PM₁BC), from the COSMO-ART application (case 2; green line) and the measurements (black dots). The RWB fraction (%) of black carbon (BC_wb), as predicted (case 2) and observed is shown by the grey and the black line, respectively (right axis). The cold nights (intense RWB smog episodes) and the weekends/holidays are shown with the red and yellow line, respectively. The shadowed areas indicate: the rainy events (in light grey) and the dust event (in dark grey).





**Figure 5: The mean daily cycle of mass concentrations (μg m⁻³) from measurements (black dots) and from model outputs (case 1 in grey columns and case 2 in green columns) and of RWB fractions (%) from measurements (black line; Only BC$_{wb}$ fraction is measured) and from model outputs (case 2 in green line) for: (a) PM$_{10}$, (b) PM$_1$OA, and (c) PM$_1$BC. All values refer to the site of Thissio (Athens), and they correspond to the intense smog wintertime period.**





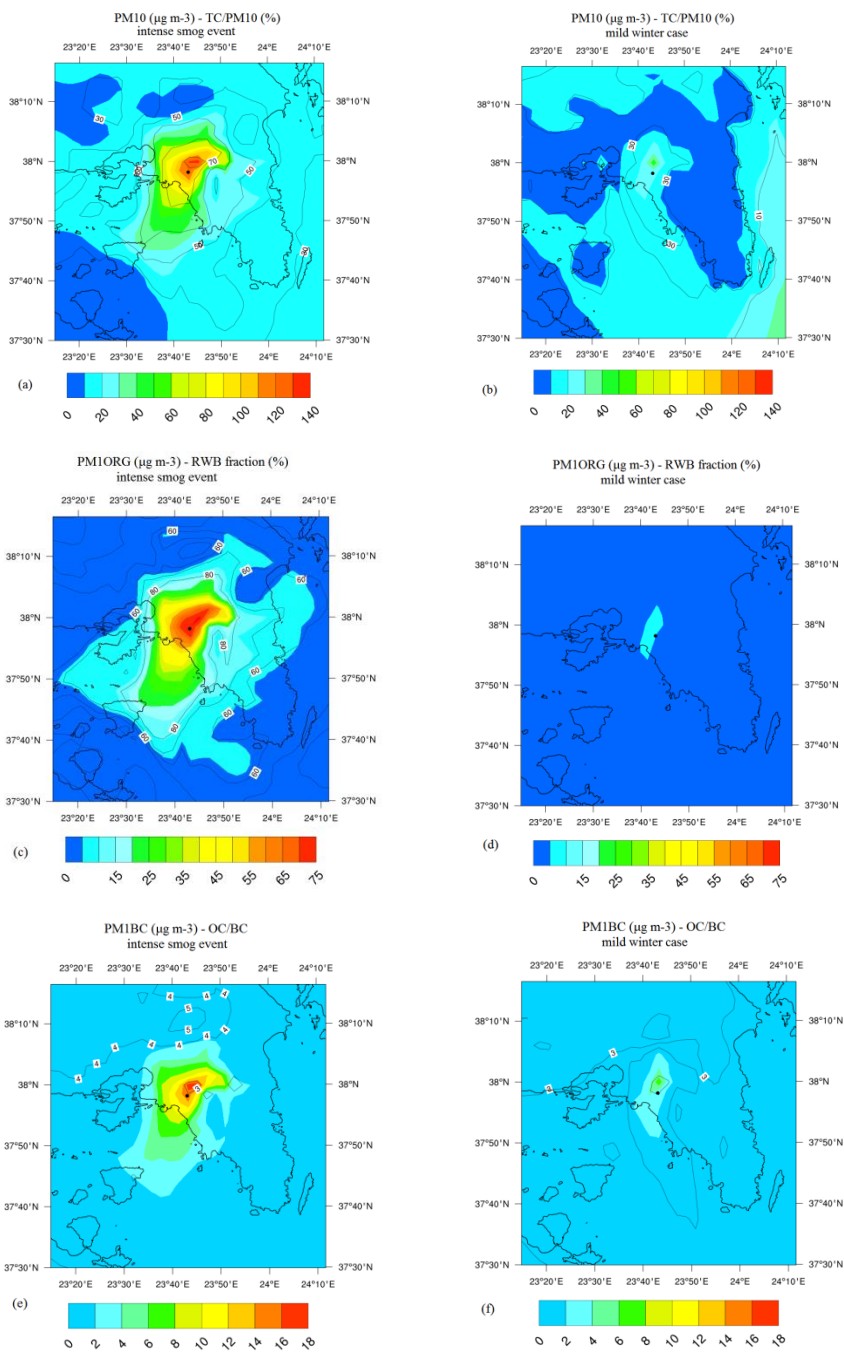

**Figure 6: Mean daily surface concentration (µg m$^{-3}$) fields of: (a) and (b) PM$_{10}$. Isolines represent the carbonaceous fraction (OA + BC) of PM$_{10}$ (%), (c) and (d) PM$_1$OA. Isolines represent the RWB fraction of PM$_1$OA (%), (e) and (f) PM$_1$BC. Isolines represent the OC/BC ratio. OC is approximated as OA divided by 1.6, a ratio suggested by Turpin and Lim (2001) for urban areas. All maps cover the extended area of Athens (Thissio is marked with the black dot) and derive from model outputs that correspond to the intense smog event (case 1; left column) and to the mild winter case (case 3; right column).**

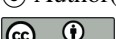



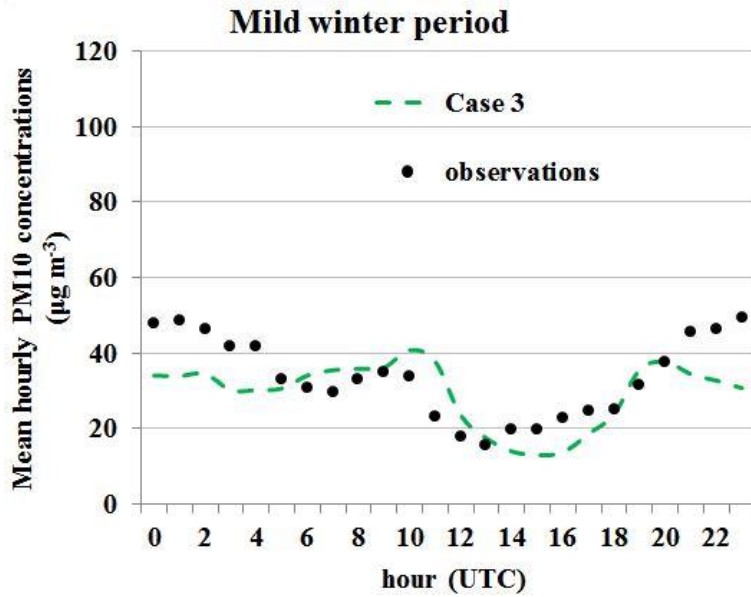

**Figure 7:** The linear correlation between air temperature (°C) and submicron organic aerosol concentration peaks (PM$_1$OA in μg m$^{-3}$, in black) and between the heating demand (HD, in °C) and the model bias (for PM$_1$OA in μg m$^{-3}$, in green), (b) the mean daily cycle of PM$_{10}$ concentrations during the mild winter period, from the observations (black dots) and model predictions by case 3 (green line). All data refer to the Thissio site (Athens).

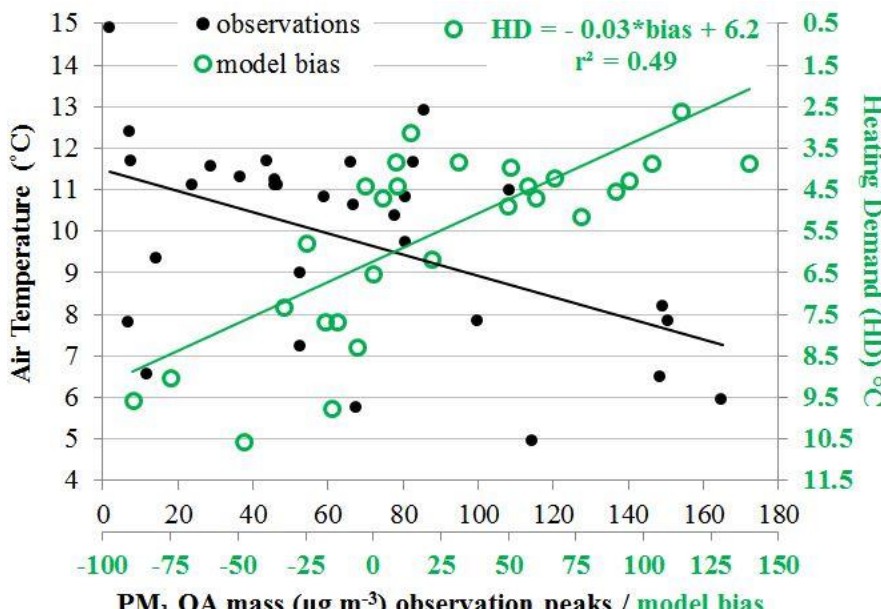

**Figure 8:** The linear correlation between air temperature (°C) and submicron organic aerosol concentration peaks (PM$_1$OA in μg m$^{-3}$, in black) and between the heating demand (HD, in °C) and the model bias (for PM$_1$OA in μg m$^{-3}$, in greenAll data refer to the Thissio site (Athens).





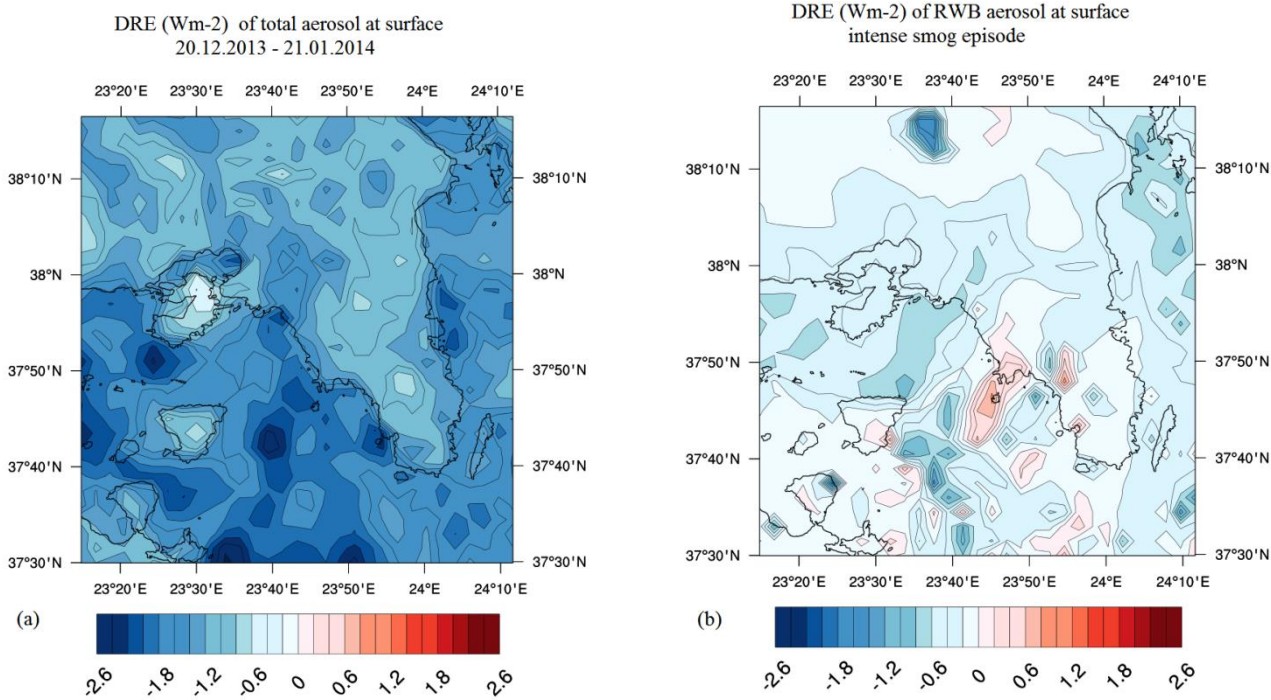

**Figure 9: The spatial distribution of the direct radiative effect (Wm$^{-2}$) of the aerosol load at the ground level, over the extended area of Athens, during: (a) a winter month (20 December, 2013 – 21 January, 2014), taking into account the total aerosol load (mean monthly difference between case 2 and case 4) and (b) the RWB smog episode (4 – 5 January, 2014), isolating the RWB aerosol load (mean daily difference between case 2 and case 3).**