# Peer review of "Changes in the domestic heating fuel in Greece: effects on atmospheric chemistry and radiation"

_Atmospheric Chemistry and Physics, 2017_

## Referee Comment (RC1) · Anonymous Referee #2 · 12 Apr 2017

General comments:

The manuscript concerns the impact of increasing burning for residential heating in Athens, Greece.

Even if the Greek situation is peculiar due to the impact of the economic crisis, this subject has great importance over the whole Europe, where the use of wood burning for house heating is increasing in a number of areas of different states. Even if the scientific knowledge of the dangers associated to biomass burning is growing, the air quality impact of biomass burning in domestic devices is not yet perceived and understood by the population and decision makers.

The manuscript is properly organized and well written. It points out that biomass burning for house heating has a major impact on the air quality and on atmospheric composition, while it has a minor influence on local radiative forcing.

The authors describe their approach to improve wood burning description within the emission modelling, that could be useful for model development and application in different areas and contexts.

Because the manuscript is proposed for publication in a special issue on coupled chemistry-meteorology modelling, discussion on the relevance of the on-line coupling approach on the presented results concerning atmospheric composition and aerosol chemistry is presently missing. The on-line coupling advantage is evident only for the studied aerosols feedback on meteorology, while it should be specified is any difference could be noticed on pollutants concentrations when feedback was switched off (Scenario 4).

The present form of the manuscript needs a revision including: clarifications, figures improvement, extension of feedback effects discussion.

Specific comments:

2.2 Model framework and setup

Line 3

The model domain is defined to be "the extended area of Greece" (the same definition is repeated in Table 1). This definition is quite generic and should be made more specific adding a Figure or a better definition of the domain boundaries.

Lines 5-6

The sentence "The atmospheric pressure...." is not understandable in this form. How where pressure and precipitation optimized? Do the authors refer to the choice of the

microphysics scheme? What is the mentioned optimization?

Line 9

Does "constant initial conditions" mean uniform initial conditions?

Table 1

It is not clear how the initial and boundary conditions for aerosols are defined. Are the values included in Table 1 uniform in space? Are those values kept constant at boundaries?

2.3 Modifications of the aerosol emissions

Figure 2 is hardly understandable. Its quality should be improved.

Lines 21-23

Does the sentence "Combined with the temporal..." refer to Figure 2?

Does Figure 2 describe average emissions or do plotted values refer to a specific hour?

Lines 23-24

Maximum wood burning emissions are said to be located at the urban core, while it would be reasonable to expect to have maximum emissions over peripheral areas, where the access to wood should be easier that in the center.

2.4 The aerosol optical properties

Line 9

Concerning aerosols composition, it is not clear how the concentration values reported in brackets should be interpreted.

Lines 15-18

The values used for Athens differ from those used by Vogel et al. (2009). Is the difference due to the geographic area of application or to any other understandable reason?

To which geographic region do values reported by Takemura et al. refer?

3.1 Impacts of residential wood burning (RWB) on atmospheric aerosol mass and chemistry

3.1.1 Aerosol model performance under smog influence

Line 23

Do values reported in Table 3 as "daily mean" refer to the whole period mean as understood from the manuscript text?

Lines 1-2

How (on the basis of what parameters) is it evaluated the mentioned 70% improvement?

Lines 10-11

The meaning of the sentence "which leads to the improvement of the half PM1 OA and of all PM1 BC the daytime peaks during the intense smog period" is not clear.

3.1.2 Representative spatial aerosol fields

Lines 16-17

The reference to PM10 EU alarm threshold is not clear, please include proper references to EU directives.

3.2 Impact of RWB smog on radiation

Lines 25-27

The sentence concerning removal of absorbing BC is not very clear and should be rephrased to be more clearly understandable.

---

## Referee Comment (RC2) · Anonymous Referee #1 · 12 May 2017

Review of the paper "Changes in the domestic heating fuel in Greece: effects on atmospheric chemistry and radiation" by Eleni Athanasopoulou and co-authors.

This is an interesting and well written paper. In general, the approach that was followed is sound and the assumptions can be followed. There are some points that the authors need to explain in more detail. They need to better justify the approach and the model setup they have chosen. I also suggest some modifications w.r.t. the wording as well as to figures and tables.

Major comments:

In section 2.2 and table 1 you should better explain the model setup. It is not clear

to me which areas were modeled with COSMO-ART, if there is a coarser grid around Athens that was modeled with a CTM as well. It looks like the area you ran COSMO-ART on is about 30x40 grid points. This is rather small, in particular when you use boundaries for the gaseous species which are on a 2.5° x 1.9° grid. In this case one grid cell of the outer grid would be bigger than your entire model domain. In addition, the horizontal resolution of your emissions is much coarser than the model resolution, by a factor of 2.5 x 5. The question arises why you did not make any attempts to use spatial surrogates like population density to redistribute the emission data and provide them in a higher resolution as model input.

page 2, line 21 and in the following: You use BC and EC as it is the same. You need to explain this a bit, e.g. provide definitions of both of them and their relation to what you call "soot". This is of particular importance because the radiative effects of the aerosol is in close connection to the BC/EC concentrations. page 5, line5: "The atmospheric pressure and precipitation parameters were optimized with respect to the high spatial resolution of the current application (0.025ËŽ)". This could be anything, therefore you need to explain what you did with the data.

page 5, line 16: If I understand it correctly, you model 2013/14 but you used 2009 emissions for all species and sectors, except for wood combustion. This can be well justified if no big changes in the emissions from year to year can be expected. However, you argue that "non-solid fuels" became very expensive (page 2, line 2/3) and that the financial crisis changed the emission pattern quite drastically. Shouldn't this be considered in the emissions of other sectors than wood burning as well?

page 5, line 21: "For Greece, it is calculated that almost all mass (i.e. 98%) of the total PM10 emissions from this source category reflects wood combustion (i.e. open fireplaces)": Is this given in the TNO emission inventory? Or is this given in the IIASA publication which you cite later (line 25)? You should state this more clearly.

page 5, line 31-page6, line 11: This change in the temporal profile is an important

modification of the main PM emission pattern. I think it would be good to explain in some more detail how you did that, e.g. by showing the long term BC measurements you mention. Can you be sure that the BC observations are dominated by wood combustion and not heavily influenced by traffic?

page 6, line 21: Although you modified the temporal profile of the RWB emissions you did not improve the spatial distribution, e.g. by using higher resolved population density maps. Why not? Shouldn't this make a significant difference? Did you make an attempt to couple the day-by-day variation of the emissions to the ambient temperature?

page 6, line 26-31: I didn't get what the assumptions are concerning the magnitude of the emissions. If I understand you correctly, you took the emitted PM10 mass from the TNO inventory without further changes, although you explained before that the assumptions about the fuel split made in this inventory are not valid for the time period you investigate here. In the end, you argue that the magnitude of the emissions is ok because the agreement between model results and observations indicates that. This seems to be inconsistent.

page 6, line 15: What is the reason for the differences between your Mie calculations and the values used in COSMO-ART? Is the conclusion that the optical properties for soot are incorrect in COSMO-ART?

page 6, line 20: "The period studied can be characterized as a relatively mild winter period.": Why did you choose this period when you can expect that the heating activities will be comparably low?

page 8, line 24/25: Is the intense smog period the entire period from 19 Dec to 21 Jan? The mean concentration values for case 2 agree better to the observations than those from the baseline although the total emission amount was the same. Is that correct? You should make this clear.

page 9, line 12 and line 15: You should avoid descriptions like "satisfactorily repre-

sented" and "nicely captured". Each reader might think differently about what is satis-fying or nice. Obviously, for case 2 the temporal profile of the concentrations fits better to the observations than for case 1 and you should just describe this. You could give a correlation coefficient to show this in numbers.

page 10, line 33: "The peaks observed at the urban core, as revealed from this model application, are somewhat displaced from the exact location of the site. This finding is in line with the characterization . . .": This is a bit over-interpreted, having the coarse emission map in mind. The grid cells with the highest emissions are simply north of Thissio.

page 12, line 9/10: " . . . for a pre-crisis period or significant change in dominant heat-ing fuels, the TNO-MACC_II emissions from residential heating should be further ad-justed.": Why for a pre-crisis period? Didn't you say that the fuels changed significantly during the crisis? Why is the TNO inventory then wrong in a pre-crisis period?

page 14: I do not agree with the last part of the conclusions. In particular: - "Thus, human health implications, as well as policy making, when the fireplaces are in use to cope with high heating demand conditions (HD > 7.5 °C), can be satisfactorily esti-mated and planned with the aid of such a tool.": You did not estimate any human health effects with your model system. How will help to estimate them? What can be planned with the help of the model results? - "For mild winter conditions (Tmin > 8-9 °C), a post-processing of model results according to the linear regression between HD and model bias, can further improve the quality of the model system.": Wouldn't it be better to improve the emission estimates than to correct the model? - "Alternatively, an inter-active treatment of RWB emissions, i.e. their online adjustment according to the actual temperature conditions of the simulation period, is proposed as a means to further en-hance the reliability of operational forecasts of online-coupled atmospheric models." I agree to this, but how would you know how much wood is used in comparison to oil or other fuels?

Minor comments

page 1, line 17: "... accurately predicts ...": Qualifiers like "accurately" are always a bit difficult if do not give numbers for the deviations.

page 2, line 2: "exorbitant" should be replaced by "very high".

page 3, line 7: Denier van der Gon

page 2, line 21 and in the following: You use BC and EC as it is the same. You need to explain this a bit, e.g. provide definitions of both of them and their relation to what you call "soot"

page 3, line 32: " ... which was, however, nonlinear.": In which way nonlinear?

page 8, line2: "mean maximum nighttime PM10": over which period was the mean taken? Was this hourly?

page 9, line 1: "Overall, the revised run improved more than the 70% of the day and nighttime PM10 peaks during the intense smog period.": This is not well formulated and therefore unclear. What exactly is improved?

page 9, line 10/11: "which leads to the improvement of the half PM1OA and of all PM1BC the daytime peaks during the intense smog period." This sentence is obscure.

page 9, line 28-30: "Thus, the aerosol chemical composition during the economic crisis is completely altered with respect to the chemical profile of wintertime aerosols beforehand." What was the chemical profile before the economic crisis?

page 10, line 2/3: "The comparison with available measurements during increased wood burning in the Alpine area (Szidat et al., 2009) reveals similarities": Where are these measurements shown? Are they given as the observations in Table 3? Please add the values they found somewhere.

page 10, line 27: "PM1BC is found below 8": The unit is missing.

page 12, line 2/3: "Inversely, during the cold days (Tmin below 8 °C) that the HD is increased (>7.5 °C), and consequently no model overestimation occurs." Omit the word "that". In addition: It is true, that no overestimation occurs. Instead, you see an underestimation.

Table 3 is really hard to read. It looks unstructured I would suggest to split into two or three different tables. One suggestion could be to have the ratios in the bottom (OC/BC, …) in separate table and also the peak analysis in another separate one. For the ratios in the bottom, it is unclear how they are given. OC/BC = 2.8 makes some sense, but what about the other values (BC/TC, …)? Are they given in %? BC/TC = 28 makes no sense. In addition, it seems that you derived them from case2/case 3 differences but you do not say this in the caption or anywhere else in the table itself.

Figure 7: The caption is wrong. It contains parts that belong to Fig. 8.

[Figure]

---

## Author Comment (AC1) · 27 Jun 2017

29. Because the manuscript is proposed for publication in a special issue on coupled chemistry-meteorology modelling, a discussion on the relevance of the on-line coupling approach on the presented results concerning atmospheric composition and aerosol chemistry is presently missing. The on-line coupling advantage is evident only for the studied aerosols feedback on meteorology, while it should be specified is any difference could be noticed on pollutants concentrations when feedback was switched off (Scenario 4). Author's response: We thank the reviewer for the positive comments and the interesting suggestion. We have calculated the differences on PM10 concentrations (and their chemical composition) between Case2 and Case4. The feedback of the online coupling approach on aerosol is found very small, which was expected given the small negative effect of particles on radiation in our case study. The section 3.2 is now named after "Impact of RWB smog on radiation and feedbacks on atmospheric composition", and a relevant discussion is now added therein (p. 13 lines 21-26), plus some additions in abstract (p. 1 line 24) and conclusions (p. 14 lines 32-33).

The present form of the manuscript needs a revision including: clarifications, figures improvement, extension of feedback effects discussion. Specific comments:

2.2 Model framework and setup Page 5 Line 3 30. The model domain is defined to be "the extended area of Greece" (the same definition is repeated in Table 1). This definition is quite generic and should be made more specific adding a Figure or a better definition of the domain boundaries. Author's response: the definition of the horizontal domain in degrees is now added in Table 1.

Lines 5-6 31. The sentence "The atmospheric pressure...." is not understandable in this form. How where pressure and precipitation optimized? Do the authors refer to the choice of the microphysics scheme? What is the mentioned optimization? Author's response: In the COSMO-ART (and pre-processor) configuration, there are certain parameters/flags that are spatially-sensitive. Among others, these settings correspond to a better balancing of the pressure fields and to a tuning in precipitation, in accordance to the horizontal spatial resolution of 0.025 deg. Therefore, the input data remained intact, the microphysics scheme was not altered, but e.g. the mask for smoothing of steep orography is adjusted, the critical value for normalized over-saturation is adjusted and so on. It should be noted that these changes are imposed either by using 'true' or 'false' or by applying certain code numbering in specific flags of the model (and pre-processor) job scripts. In order to avoid further puzzling of readers, we have decided to remove the respective phrase from the document.

Line 9 32. Does "constant initial conditions" mean uniform initial conditions? Author's

[Figure]

response: Combining this with the next comment (no. 33), the phrase (p. 5 line 14) is now replaced by: "...the uniform (in space) and constant (in time) initial conditions (e.g. for SO2 and aerosol species)."

Table 1 33. It is not clear how the initial and boundary conditions for aerosols are defined. Are the values included in Table 1 uniform in space? Are those values kept constant at boundaries? Author's response: Yes to both. This is now better explained in text (cf. response to Comment no. 32).

2.3 Modifications of the aerosol emissions 34. Figure 2 is hardly understandable. Its quality should be improved. Author's response: all figures will be uploaded separately in high quality, following the ACP guidelines.

Page 6 Lines 21-23 35. Does the sentence "Combined with the temporal..." refer to Figure 2? Author's response: After this comment, the phrase (p. 7 lines 3-4) is modified for clarity as: "...in Figure 2. These rates, combined with the temporal profiles (Figure 1),..."

36. Does Figure 2 describe average emissions or do plotted values refer to a specific hour? Author's response: These are hourly emissions "for a night hour of a weekday (Tuesday, 21.00 UTC)", as indicated in the caption of the figure.

Lines 23-24 37. Maximum wood burning emissions are said to be located at the urban core, while it would be reasonable to expect to have maximum emissions over peripheral areas, where the access to wood should be easier that in the center. Author's response: In principle, population density is used as a spatial proxy in order to distribute wood combustion emissions (TNO report, 2010). The TNO-MACC_II emission database we have used for this study (Kuenen et al., 2014), uses a population map at high resolution, and a special proxy for the distribution of residential wood combustion. The latter takes into account both the population density, but also the proximity to wood (for more information on proxies for RWB, cf. Comment no. 7). Despite this combination for the distribution of residential wood combustion, an overallocation of the

emissions in urbanized centres is possible, as the reviewer also comments, which is already described in previous studies (Denier van der Gon et al., 2014 and Timmermans et al., 2013). We have now included this info in the text. (p. 7 lines 5-10).

2.4 The aerosol optical properties Page 7 Line 9 38. Concerning aerosols composition, it is not clear how the concentration values reported in brackets should be interpreted. Author's response: These are RWB smog period-averages of the respective hourly values measured at Thissio during the RWB periods of winter 2013-14. This is explained in the text, in the previous sentence: "...is based on observational data collected in Greece during the 2013-2014 RWB smog episodes. In particular, the average surface chemical composition of ultrafine aerosols in Athens (pure soot: 2.8 $\mu$g m-3, water soluble mixture of sulfate, nitrate, ammonium and organics: 22.2 $\mu$g m-3), local relative humidity (50-70 %) and an average mixing layer height (600 m a.g.l., Gerasopoulos et al., 2017), were used to feed the OPAC software (Hess et al., 1998), which then - by applying the Mie theory - provides the respective optical properties for 61 wavelengths between 0.25 and 40 $\mu$m". Thus, no interpretation should be extracted from these values; we provide information on the representative (average) values we used to calculate the aerosol optical properties for our case study.

Lines 15-18 39. The values used for Athens differ from those used by Vogel et al. (2009). Is the difference due to the geographic area of application or to any other understandable reason? Author's response: Yes, the difference is related both to the geographic area and to the selected events. The phrase (p. 7 lines 31-32) is now enriched as follows: "These values differ from the ones used in Vogel et al. (2009; black and red lines in Figure 3), which is expected due to the different geographical areas and periods of interest between the two studies."

40. To which geographic region do values reported by Takemura et al. refer? Author's response: They do not correspond to a specific area, but are used as model inputs for the model study of Takemura et al. (2002). Thus, they are directly comparable to our findings. This is now more clearly explained in p. 8 line 2).

3.1 Impacts of residential wood burning (RWB) on atmospheric aerosol mass and chemistry 3.1.1 Aerosol model performance under smog influence Page 8 Line 23 41. Do values reported in Table 3 as "daily mean" refer to the whole period mean as understood from the manuscript text? Author's response: Yes, these are averaged daily mean values for the RWB smog sub-period of the simulation period. The phrase 'averaged daily mean' is now used instead of 'daily mean'.

Page 9 Lines 1-2 42. How (on the basis of what parameters) is it evaluated the mentioned 70% improvement? Author's response: The relevant phrase is now written as "Overall, the revised run shows improvements in the calculated PPEA values for more than the 70% of the day and nighttime PM10 peaks during the intense smog period.", so that this question is answered in the manuscript (p. 9 lines 25-26).

Lines 10-11 43. The meaning of the sentence "which leads to the improvement of the half PM1 OA and of all PM1 BC the daytime peaks during the intense smog period" is not clear. Author's response: The relevant phrase is now written as "The PPEA values for both carbonaceous species are significantly lower for case 2, i.e. for the half PM1OA and of all PM1BC the daytime peaks during the intense smog period", so that its meaning is clear (p. 9 line 33 – p. 10 line 2).

3.1.2 Representative spatial aerosol fields Page 10 Lines 16-17 44. The reference to PM10 EU alarm threshold is not clear, please include proper references to EU directives. Author's response: The reviewer is correct. The PM10 alarm value we refer to, is not suggested by any EU directive, but rather by the National Legislation (Joint Ministerial Decision by FEK 3272ÎŠ/23-12-2013, in Greek), which supplements the respective EU directive with respect to public information and emission reduction thresholds. The acronym 'EU' is now replaced with 'National' within text and a proper reference is added (p. 11 line 9). The same changes are now done for the alert threshold for the whole population (p. 6 line 29).

3.2 Impact of RWB smog on radiation Page 12 Lines 25-27 45. The sentence

concerning removal of absorbing BC is not very clear and should be rephrased to be more clearly understandable. Author's response: This sentence (p. 13 lines 18-20) is now rephrased as: "...while the daytime mean was 98 $\mu$g m-3, 50% of which corresponding to RWB particles. By further subtracting the absorbing BC aerosols (14 $\mu$g m-3) from the RWB mean PM10 concentration value, it is found that..."

Please also note the supplement to this comment:
https://www.atmos-chem-phys-discuss.net/acp-2017-139/acp-2017-139-AC1-supplement.pdf

---

## Author Comment (AC2) · 27 Jun 2017

1. In section 2.2 and table 1 you should better explain the model setup. It is not clear to me which areas were modeled with COSMO-ART, if there is a coarser grid around Athens that was modeled with a CTM as well. It looks like the area you ran COSMOART on is about 30x40 grid points. This is rather small, in particular when you use boundaries for the gaseous species which are on a 2.5 x 1.9 grid. In this case one grid cell of the outer grid would be bigger than your entire model domain. In addition, the horizontal resolution of your emissions is much coarser than the model resolution, by a factor of 2.5 x 5. The question arises why you did not make any attempts to use

spatial surrogates like population density to redistribute the emission data and provide them in a higher resolution as model input. Author's response: There is a single area modeled with COSMO-ART, "the extended area of Greece, centered on Athens" (p.5 line 8). The covered area is 18 to 30 $E^\circ$ and 33 to 42 $N^\circ$. The horizontal spatial resolution is 0.025 deg, thus the whole domain is composed by 384 x 330 grid points. The reviewer though is correct, the horizontal extend of the domain was not explicitly mentioned in the initial version of the paper. It is now added in Table 1. With respect to the grid differences between emission data and model runs, first please note that the size factor between the TNO/MACC inventory and the COSMO-ART grid is smaller than a factor 2.5 x 5, since COSMO-ART ran on a rotated grid with equator placed over Greece. At the latitude of Greece, the size factor is in the order of 2.5 x 3.9. Still, emission downscaling could be beneficial, but this has not yet been implemented in the official COSMO-ART emission pre-processor. A proper downscaling is a quite complex issue (see e.g. Lopez-Aparicio et al., Atmos. Environ. 154, 285-296, 2017). For instance, population data alone is not sufficient for this purpose, since it is a poor proxy for industrial sources or power plant emissions and only a moderately reliable proxy for traffic. With respect to the suitable proxy(-ies) for residential wood combustion, see Comments no. 7 and 37.

2. page 2, line 21 and in the following: You use BC and EC as it is the same. You need to explain this a bit, e.g. provide definitions of both of them and their relation to what you call "soot". This is of particular importance because the radiative effects of the aerosol is in close connection to the BC/EC concentrations. Author's response: The reviewer is of course correct about the different definitions of black carbon, elemental carbon and soot. Apart from this specific statement in the Introductory section, we only deal with BC data throughout the current study. Therefore, we see no actual need for the provision of the definitions and inter-relation, rather than providing a relevant reference and a statement with respect to their erroneous usage as synonyms (p. 2 line 22).

3. page 5, line5: "The atmospheric pressure and precipitation parameters were optimized with respect to the high spatial resolution of the current application (0.025ËŽ)". This could be anything, therefore you need to explain what you did with the data. Author's response: In the COSMO-ART (and pre-processor) configuration, there are certain parameters/flags that are spatially-sensitive. Among others, these settings correspond to a better balancing of the pressure fields and to a tuning in precipitation, in accordance to the horizontal spatial resolution of 0.025 deg. Therefore, the input data remained intact, the microphysics scheme was not altered, but e.g. the mask for smoothing of steep orography is adjusted, the critical value for normalized over-saturation is adjusted and so on. It should be noted that these changes are imposed either by using 'true' or 'false' or by applying certain code numbering in specific flags of the model (and pre-processor) job scripts. In order to avoid a similar confusion by the readers, we suggest deleting the respective phrase from the document.

4. page 5, line 16: If I understand it correctly, you model 2013/14 but you used 2009 emissions for all species and sectors, except for wood combustion. This can be well justified if no big changes in the emissions from year to year can be expected. However, you argue that "non-solid fuels" became very expensive (page 2, line 2/3) and that the financial crisis changed the emission pattern quite drastically. Shouldn't this be considered in the emissions of other sectors than wood burning as well? Author's response: In principle, the reviewer is correct, the latest available data relevant to emission rates should be considered for atmospheric modelling. Nevertheless, the latest available, official emission data from the TNO-MACC_II database refer to year 2009, which we were able to update according to recent specialized measurements of aerosol species during recent, intensive RWB wintertime periods. A recent study on air pollution emissions in Greece based on actual data (Fameli et al., 2016) shows that with respect to residential heating, fuel consumption is more or less stable for all fuels from 2009 to 2012 (latest available data), except for wood which is increased. With respect to the road sector, we cannot estimate the impact of crisis on transport, since available data end in 2010. This is now added in the text (p. 5 lines 23-26)

5. page 5, line 21: "For Greece, it is calculated that almost all mass (i.e. 98%) of the total PM10 emissions from this source category reflects wood combustion (i.e. open fireplaces)": Is this given in the TNO emission inventory? Or is this given in the IIASA publication which you cite later (line 25)? You should state this more clearly. Author's response: Yes, this is a finding based on the TNO emission data for Greece, by comparing PM10 total residential combustion to PM10 residential wood burning (mass) emissions (personal communication with Hugo Denier van der Gon). It is true though that the TNO_MACC-II methodology incorporates model approximations, when data for certain sectors/countries are unavailable. In particular, reported data for PM10 and PM2.5 emissions were not available for Greece, thus emissions were taken from the GAINS model (IIASA, 2012). This is now more clearly stated in the document (p. 6 line 6).

6. page 5, line 31-page6, line 11: This change in the temporal profile is an important modification of the main PM emission pattern. I think it would be good to explain in some more detail how you did that, e.g. by showing the long term BC measurements you mention. Can you be sure that the BC observations are dominated by wood combustion and not heavily influenced by traffic? Author's response: It is true that the revised temporal cycles led to crucial changes in the daily and weekly cycles of the PM concentrations (e.g. Figure 5). This revision was based solely on the wood burning fraction of BC (BCwb; method of separation of BC to BCwb and BCff -fossil fuel- as already described in Sect. 2.1), thus not influenced by traffic or other sources. Second, both temporal cycles correspond to the mean values of the long-term measurements (2013-2015) of BCwb in Athens, normalized with respect to the average value of each cycle. So actually Figure 1 reflects directly the long-term (normalized) measurements of BCwb, as suggested by the reviewer's comment. This is analytically described in Sect. 2.3, but now also added in the caption of Figure 1.

7. page 6, line 21: Although you modified the temporal profile of the RWB emissions you did not improve the spatial distribution, e.g. by using higher resolved population

density maps. Why not? Shouldn't this make a significant difference? Did you make an attempt to couple the day-by-day variation of the emissions to the ambient temperature? Author's response: As already stated in Comment no. 1, emission downscaling has not yet been implemented in the official COSMO-ART emission pre-processor. Using proxies for wood burning is even more elusive. Discussions are under way at the European level (e.g. FAIRMODE forum) for the best approach on distributing wood burning emissions. The geographical spread of fire places/wood stoves use, would probably be a better proxy than the population density for the estimation of RWB emissions, which are two parameters not necessarily inter-correlated. Such a coupling went beyond the scope of this study, but is a currently ongoing effort of our group, so that the downscaling of residential combustion emissions in the Athens basin to the intra-urban scale is performed, based on construction and socio-economic criteria. With respect to the correlation of air pollution to temperature, this is already performed for PM concentrations, shown in Figure 4 (cold nights are highlighted) and Figure 8 (OA vs. Tmin). It is explicitly written in the text that this correlation reflects the coupling of the day-by-day variation of the emissions to the ambient temperature (so called 'heating demand'; p. 12 lines 30-32). We have concluded that the performance of their numerical coupling online can be an added value for the model system, when long-term winter-time model outputs are necessary, during the current period (intense use of fireplaces). Nevertheless, if one wishes to focus on RWB smog events, as in our case, coupling is not crucial, as model outputs capture the measured aerosol peaks.

8. page 6, line 26-31: I didn't get what the assumptions are concerning the magnitude of the emissions. If I understand you correctly, you took the emitted PM10 mass from the TNO inventory without further changes, although you explained before that the assumptions about the fuel split made in this inventory are not valid for the time period you investigate here. In the end, you argue that the magnitude of the emissions is ok because the agreement between model results and observations indicates that. This seems to be inconsistent. Author's response: Indeed, the total PM10 mass emitted from the residential combustion section, provided by the TNO database, was

not altered, as on a daily basis there was good agreement between model results and measurements of total PM10, especially during cold days/smog events. However, the carbonaceous PM components (OA and BC) where poorly captured. Thus, our improvements involved a different chemical profile to the residential combustion aerosol emissions (different emitted mass of OA and BC component of PM10) than the TNO (central heating profile), driven by the respective, specialized (RWB) measurements (p. 6 line 27- page 7 line 2).

9. page 6, line 15: What is the reason for the differences between your Mie calculations and the values used in COSMO-ART? Is the conclusion that the optical properties for soot are incorrect in COSMO-ART? Author's response: No, the optical properties for aerosol particles incorporated in COSMO-ART are correct. Nevertheless, they have been defined for typical air pollution conditions in an area of Germany. Our calculations are based on local conditions with respect to the aerosol chemical profile for the event/area of interest, as well as for the local, representative RH and PBL height (cf. Sect. 2.4). Thus, they differ from the ones in COSMO-ART, but they are similar both to other findings in the Mediterranean (Mishra et al., 2014; 2015), as well as to respective model inputs (Takemura et al., 2002). The relevant phrase is now enriched as follows: "These values differ from the ones used in Vogel et al. (2009; black and red lines in Figure 3), which is expected due to the different geographical areas and periods of interest between the two studies." (p.7 lines 31-32).

10. page 6, line 20: "The period studied can be characterized as a relatively mild winter period.": Why did you choose this period when you can expect that the heating activities will be comparably low? Author's response: Indeed, the winter of 2013-14 was mild. Nevertheless, the period we have focused our research on the RWB implications on air pollution issues (19 Dec – 5 Jan) can be characterised as an intense smog period, as explained in p. 8 lines 18-23). We have rephrased the initial statement (p. 8 line 5), so that the reader is not confused with respect to the whole winter and the selected smog period.

11. page 8, line 24/25: Is the intense smog period the entire period from 19 Dec to 21 Jan? The mean concentration values for case 2 agree better to the observations than those from the baseline although the total emission amount was the same. Is that correct? You should make this clear. Author's response: No, the smog period is not the entire modelled period. Taken from the text (p. 8 lines 18-20): "19 December 2013 – 05 January 2014 . . . excluding the rainy days mentioned above, constitutes the intense smog period of the current modeling study". Yes, the reviewer is correct, a clarification and explanation is needed for this finding. In fact, although the total PM10 mass emitted by residential combustion is unchanged within a typical week, the weekday mass amounts are reduced due to the revised weekly cycles, which are instead emitted during the weekends (cf. Fig. 1b). Given the fact that the intense smog period (13 days) includes only 4.5 weekend days, the differences found by the two scenarios in the average PM10 concentrations are expected. This is now explained in the revised text (p.9 lines 13-16).

12. page 9, line 12 and line 15: You should avoid descriptions like "satisfactorily represented" and "nicely captured". Each reader might think differently about what is satisfying or nice. Obviously, for case 2 the temporal profile of the concentrations fits better to the observations than for case 1 and you should just describe this. You could give a correlation coefficient to show this in numbers. Author's response: 'satisfactorily represented' is replaced by 'reproduced' and 'nicely' has been removed. Correlation coefficients are added for the daily cycles of carbonaceous species, and mean hourly bias is added in the peak comparisons (p. 10 lines 3-4, lines 6-8).

13. page 10, line 33: "The peaks observed at the urban core, as revealed from this model application, are somewhat displaced from the exact location of the site. This finding is in line with the characterization. . .": This is a bit over-interpreted, having the coarse emission map in mind. The grid cells with the highest emissions are simply north of Thissio. Author's response: We agree over possible over-interpretation, which is why we have calculated the mean, minimum and maximum values of a greater urban

area (118 km2, 15 cells; SW corner: 37.94ËŽ, 23.67ËŽ) and compared them to the model results at the grid point of Thissio. The results, which are given in p. 11 line 30 – p. 12 line 5), further support the argument already posed by Gratsea et al. (2016), on the characterization of Thissio as an urban background site, not intensively affected by local traffic and representative of the average background pollution conditions in Athens.

14. page 12, line 9/10: "... for a pre-crisis period or significant change in dominant heating fuels, the TNO-MACC_II emissions from residential heating should be further adjusted.": Why for a pre-crisis period? Didn't you say that the fuels changed significantly during the crisis? Why is the TNO inventory then wrong in a pre-crisis period? Author's response: It is true that during the pre-crisis period, Greek households did not use wood as the primary heating fuel, but central heating installations, which emit far less PM10 mass than wood combustion. As evident from our study, TNO-MACC_II PM10 emission rates for Greece, better fit to this RWB increase, rather than the central heating emission conditions. Thus, caution should be placed when the simulated periods reflect non-wood residential combustion, but the usage of other fuels, such as gas. The reason behind these findings might be that for the case of PM10 and PM2.5 in Greece, TNO had no data available, thus applied model approximations for the calculation of emissions (cf. reply to Comment no 5).

15. page 14: I do not agree with the last part of the conclusions. In particular: - "Thus, human health implications, as well as policy making, when the fireplaces are in use to cope with high heating demand conditions (HD > 7.5 C), can be satisfactorily estimated and planned with the aid of such a tool.": You did not estimate any human health effects with your model system. How will help to estimate them? What can be planned with the help of the model results? - "For mild winter conditions (Tmin > 8-9 C), a post-processing of model results according to the linear regression between HD and model bias, can further improve the quality of the model system.": Wouldn't it be better to improve the emission estimates than to correct the model? - "Alternatively, an interactive treatment of RWB emissions, i.e. their online adjustment according to the actual temperature conditions of the simulation period, is proposed as a means to further enhance the reliability of operational forecasts of online-coupled atmospheric models." I agree to this, but how would you know how much wood is used in comparison to oil or other fuels? Author's response: The reviewer is correct; we are not estimating health effects within this study. Nevertheless, these or similar model results can be used to provide maps with exceedances of the EU limits of PM, thus to estimate the differential exposure of population to air pollution threats. Such information could be of help to policy makers as well, with respect to warnings and measures against RWB emissions and population exposure during wintertime. Yes, we agree that the best approximation would be to improve emission estimates (online) than correct the model outputs (offline). In other words, a hybrid model of RWB treatment of emissions during the model runs, could be beneficial, depending on the scope of the study. The proposed methodology does not need the information with respect to the use of wood in comparison to other fuels, because it will adjust the given TNO mass emissions according to the ambient temperature (timestep or hourly model values), based on actual measurements (PM vs. HD) that already include this behavioural response to the use of the different heating fuels. Such a modification is out of the scope of the current study, but valuable in future studies, targeting long-term wintertime periods or operational forecasts rather than intense and specific smog episodes. A supporting statement to this, is now added in p. 12 line 30).

Minor comments 16. page 1, line 17: ": : : accurately predicts : : :": Qualifiers like "accurately" are always a bit difficult if do not give numbers for the deviations. Author's response: 'accurately predicts' is replaced by 'reproduces the measured'

17. page 2, line 2: "exorbitant" should be replaced by "very high". Author's response: done

18. page 3, line 7: Denier van der Gon Author's response: done

19. page 2, line 21 and in the following: You use BC and EC as it is the same. You need to explain this a bit, e.g. provide definitions of both of them and their relation to what you call "soot" Author's response: The reviewer is of course correct about the different definitions of black carbon, elemental carbon and soot. Apart from this specific statement in the Introductory section, we only deal with BC data throughout the current study. Therefore, we see no actual need for the provision of the definitions and inter-relation, rather than providing a relevant reference and a statement with respect to their erroneous usage as synonyms (p. 2 line 22).

20. page 3, line 32: " : : : which was, however, nonlinear.": In which way nonlinear? Author's response: The ratio of $\Delta F$ (net radiative flux in W/m-2) and AOD (at 450 nm), has a linear relation for the net shortwave flux at the surface, i.e. the points in a scatter plot (y-axis $\Delta F$, x-axis AOD) form a line with a negative slope (Stanelle et al., 2010). For the longwave the graph is not a line and resembles more a logarithmic graph. The text is now written as: "Stanelle et. al. (2010) concentrated on dust episodes over West Africa and found an average increase of 70 Wm-2 for the long-wave radiative effect. Comparing the AOD (at 450nm) with the surface long-wave flux, he found the relationship to be nonlinear in contrast to the shortwave case." (p. 33 - p. 4 line 2)

21. page 8, line2: "mean maximum nighttime PM10": over which period was the mean taken? Was this hourly? Author's response: the period is 19 December 2013 – 05 January 2014. Yes, this is an hourly value. The whole phrase now is "The meteoro-logical conditions favored the accumulation of smog over Athens during the period 19 December 2013 – 05 January 2014 (mean maximum nighttime hourly PM10 measured concentrations of 103.7 $\mu$g m-3)." (p. 8 lines 18-19).

22. page 9, line 1: "Overall, the revised run improved more than the 70% of the day and nighttime PM10 peaks during the intense smog period.": This is not well formulated and therefore unclear. What exactly is improved? Author's response: The reviewer is correct. The phrase is now formulated as: "Overall, the revised run shows improve-ments in the calculated PPEA values for more than the 70% of the day and nighttime

[Figure]

PM10 peaks during the intense smog period." (p. 9 lines 25-26).

page 9, line 10/11: "which leads to the improvement of the half PM1OA and of all PM1BC the daytime peaks during the intense smog period." This sentence is obscure. Author's response: it is again about better PPEA values. The phrase is formulated as "The PPEA values for both carbonaceous species are significantly lower for case 2, i.e. for the half PM1OA and of all PM1BC the daytime peaks during the intense smog period." (p. 9 line 33 – p. 10 line 2).

23. page 9, line 28-30: "Thus, the aerosol chemical composition during the economic crisis is completely altered with respect to the chemical profile of wintertime aerosols beforehand." What was the chemical profile before the economic crisis? Author's response: This is given in p. 6 lines 27-28: "...the original chemical profile of aerosol emissions from non-industrial combustion (20% BC, 40% OA and 40% others) was modified..."

24. page 10, line 2/3: "The comparison with available measurements during increased wood burning in the Alpine area (Szidat et al., 2009) reveals similarities": Where are these measurements shown? Are they given as the observations in Table 3? Please add the values they found somewhere. Author's response: The values are now added in Table 5.

25. page 10, line 27: "PM1BC is found below 8": The unit is missing. Author's response: The unit "$\mu$g m-3" is now added.

26. page 12, line 2/3: "Inversely, during the cold days (Tmin below 8 C) that the HD is increased (>7.5 C), and consequently no model overestimation occurs." Omit the word "that". In addition: It is true, that no overestimation occurs. Instead, you see an underestimation. Author's response: "that" is deleted from the text. We kept the rest of the sentence as was, since our main point in this figure and paragraph is model overestimation, i.e. that the model does not overestimate PM concentrations during smog events (high HD conditions) as during the milder winter days, without

actually implying that it systematically underestimates (model bias below -25% occurs in 4 cases).

27. Table 3 is really hard to read. It looks unstructured I would suggest to split into two or three different tables. One suggestion could be to have the ratios in the bottom (OC/BC,. . .) in separate table and also the peak analysis in another separate one. For the ratios in the bottom, it is unclear how they are given. OC/BC = 2.8 makes some sense, but what about the other values (BC/TC,. . .)? Are they given in %? BC/TC = 28 makes no sense. In addition, it seems that you derived them from case2/case 3 differences but you do not say this in the caption or anywhere else in the table itself. Author's response: The table is now split in 3, as suggested by the reviewer. All ratios are now dimensionless (pure numbers). RWB fraction predictions are the outcome of case 2 – case 3 differences. This is now explained in Table 3.

28. Figure 7: The caption is wrong. It contains parts that belong to Fig. 8. Author's response: The caption is now corrected: "The mean daily cycle of PM10 concentrations during the mild winter period, from the observations (black dots) and model predictions by case 3 (green line). All data refer to the Thissio site (Athens)."

Please also note the supplement to this comment:
https://www.atmos-chem-phys-discuss.net/acp-2017-139/acp-2017-139-AC2-supplement.pdf

**Supplement:**

Below, the final author comments on behalf of all co-authors are published. The response to the Referees is structured as: (1) comments from Referees (in italics), (2) author's response after each comment (normal fonts), (3) author's changes in manuscript, marked with track changes.

**5 Comments from Anonymous Referee #1:**

This is an interesting and well written paper. In general, the approach that was followed is sound and the assumptions can be followed. There are some points that the authors need to explain in more detail. They need to better justify the approach and the model setup they have chosen. I also suggest some modifications w.r.t. the wording as well as to figures and tables.

10

**Major comments:**

In section 2.2 and table 1 you should better explain the model setup. It is not clear to me which areas were modeled with COSMO-ART, if there is a coarser grid around Athens that was modeled with a CTM as well. It looks like the area you ran COSMOART on is about 30x40 grid points. This is rather small, in particular when you use boundaries for the gaseous species which are on a 2.5 x 1.9 grid. In this case one grid cell of the outer grid would be bigger than your entire model domain. In addition, the horizontal resolution of your emissions is much coarser than the model resolution, by a factor of 2.5 x 5. The question arises why you did not make any attempts to use spatial surrogates like population density to redistribute the emission data and provide them in a higher resolution as model input.

**20 Author's response:**

There is a single area modeled with COSMO-ART, "the extended area of Greece, centered on Athens" (p.5 line 8). The covered area is 18 to 30  $E^{\circ}$  and 33 to 42  $N^{\circ}$ . The horizontal spatial resolution is 0.025 deg, thus the whole domain is composed by 384 x 330 grid points. The reviewer though is correct, the horizontal extend of the domain was not explicitly mentioned in the initial version of the paper. It is now added in Table 1.

- 25 With respect to the grid differences between emission data and model runs, first please note that the size factor between the TNO/MACC inventory and the COSMO-ART grid is smaller than a factor 2.5 x 5, since COSMO-ART ran on a rotated grid with equator placed over Greece. At the latitude of Greece, the size factor is in the order of 2.5 x 3.9. Still, emission downscaling could be beneficial, but this has not yet been implemented in the official COSMO-ART emission pre-processor. A proper downscaling is a quite complex issue (see e.g. Lopez-
- 30 Aparicio et al., Atmos. Environ. 154, 285-296, 2017). For instance, population data alone is not sufficient for this purpose, since it is a poor proxy for industrial sources or power plant emissions and only a moderately reliable proxy for traffic. With respect to the suitable proxy(-ies) for residential wood combustion, see Comments no. 7 and 37.
- 35 2. page 2, line 21 and in the following: You use BC and EC as it is the same. You need to explain this a bit, e.g. provide definitions of both of them and their relation to what you call "soot". This is of particular importance because the radiative effects of the aerosol is in close connection to the BC/EC concentrations. Author's response:

The reviewer is of course correct about the different definitions of black carbon, elemental carbon and soot. Apart from this specific statement in the Introductory section, we only deal with BC data throughout the current study. Therefore, we see no actual need for the provision of the definitions and inter-relation, rather than providing a relevant reference and a statement with respect to their erroneous usage as synonyms (p. 2 line 22).

3. page 5, line5: "The atmospheric pressure and precipitation parameters were optimized with respect to
the high spatial resolution of the current application (0.025ËŽ)". This could be anything, therefore you need to explain what you did with the data.

**Author's response:**

In the COSMO-ART (and pre-processor) configuration, there are certain parameters/flags that are spatiallysensitive. Among others, these settings correspond to a better balancing of the pressure fields and to a tuning in precipitation, in accordance to the horizontal spatial resolution of 0.025 deg. Therefore, the input data remained integet the microphysics scheme was not altered, but e.g. the mask for smoothing of steep ergements is adjusted

- 5 intact, the microphysics scheme was not altered, but e.g. the mask for smoothing of steep orography is adjusted, the critical value for normalized over-saturation is adjusted and so on. It should be noted that these changes are imposed either by using 'true' or 'false' or by applying certain code numbering in specific flags of the model (and pre-processor) job scripts. In order to avoid a similar confusion by the readers, we suggest deleting the respective phrase from the document.
- 10
- 4. page 5, line 16: If I understand it correctly, you model 2013/14 but you used 2009 emissions for all species and sectors, except for wood combustion. This can be well justified if no big changes in the emissions from year to year can be expected. However, you argue that "non-solid fuels" became very expensive (page 2, line 2/3) and that the financial crisis changed the emission pattern quite drastically. Shouldn't this be considered in the emissions of other sectors than wood burning as well?
- 15 considered in the emissions of other sectors than wood burning as well? Author's response: In principle, the reviewer is correct, the latest available data relevant to emission rates should be considered for atmospheric modelling. Nevertheless, the latest available, official emission data from the TNO-MACC\_II database refer to year 2009, which we were able to update according to recent specialized measurements of aerosol species during recent, intensive RWB wintertime periods.
- A recent study on air pollution emissions in Greece based on actual data (Fameli et al., 2016) shows that with respect to residential heating, fuel consumption is more or less stable for all fuels from 2009 to 2012 (latest available data), except for wood which is increased. With respect to the road sector, we cannot estimate the impact of crisis on transport, since available data end in 2010. This is now added in the text (p. 5 lines 23-26)
- 25 5. page 5, line 21: "For Greece, it is calculated that almost all mass (i.e. 98%) of the total PM10 emissions from this source category reflects wood combustion (i.e. open fireplaces)": Is this given in the TNO emission inventory? Or is this given in the IIASA publication which you cite later (line 25)? You should state this more clearly.
- Author's response: Yes, this is a finding based on the TNO emission data for Greece, by comparing PM10 total residential combustion to PM10 residential wood burning (mass) emissions (personal communication with Hugo Denier van der Gon). It is true though that the TNO\_MACC-II methodology incorporates model approximations, when data for certain sectors/countries are unavailable. In particular, reported data for PM10 and PM2.5 emissions were not available for Greece, thus emissions were taken from the GAINS model (IIASA, 2012). This is now more clearly stated in the document (p. 6 line 6).
- 35
- 6. page 5, line 31-page6, line 11: This change in the temporal profile is an important modification of the main PM emission pattern. I think it would be good to explain in some more detail how you did that, e.g. by showing the long term BC measurements you mention. Can you be sure that the BC observations are dominated by wood combustion and not heavily influenced by traffic?
- 40 **Author's response:** It is true that the revised temporal cycles led to crucial changes in the daily and weekly cycles of the PM concentrations (e.g. Figure 5). This revision was based solely on the wood burning fraction of BC (BCwb; method of separation of BC to BCwb and BCff -fossil fuel- as already described in Sect. 2.1), thus not influenced by traffic or other sources. Second, both temporal cycles correspond to the mean values of the long-term measurements (2013-2015) of BCwb in Athens, normalized with respect to the average value of each
- 45 cycle. So actually Figure 1 reflects directly the long-term (normalized) measurements of BCwb, as suggested by the reviewer's comment. This is analytically described in Sect. 2.3, but now also added in the caption of Figure 1.

7. page 6, line 21: Although you modified the temporal profile of the RWB emissions you did not improve the spatial distribution, e.g. by using higher resolved population density maps. Why not? Shouldn't this make a significant difference? Did you make an attempt to couple the day-by-day variation of the emissions to the ambient temperature?

5

45

Author's response: As already stated in Comment no. 1, emission downscaling has not yet been implemented in the official COSMO-ART emission pre-processor. Using proxies for wood burning is even more elusive. Discussions are under way at the European level (e.g. FAIRMODE forum) for the best approach on distributing wood burning emissions. The geographical spread of fire places/wood stoves use, would probably be a better

- 10 proxy than the population density for the estimation of RWB emissions, which are two parameters not necessarily inter-correlated. Such a coupling went beyond the scope of this study, but is a currently ongoing effort of our group, so that the downscaling of residential combustion emissions in the Athens basin to the intra-urban scale is performed, based on construction and socio-economic criteria.
- With respect to the correlation of air pollution to temperature, this is already performed for PM concentrations,
  shown in Figure 4 (cold nights are highlighted) and Figure 8 (OA vs. Tmin). It is explicitly written in the text that this correlation reflects the coupling of the day-by-day variation of the emissions to the ambient temperature (so called 'heating demand'; p. 12 lines 30-32). We have concluded that the performance of their numerical coupling online can be an added value for the model system, when long-term winter-time model outputs are necessary, during the current period (intense use of fireplaces). Nevertheless, if one wishes to focus on RWB smog events, as in our case, coupling is not crucial, as model outputs capture the measured aerosol peaks.
- 8. page 6, line 26-31: I didn't get what the assumptions are concerning the magnitude of the emissions. If I understand you correctly, you took the emitted PM10 mass from the TNO inventory without further changes, although you explained before that the assumptions about the fuel split made in this inventory are not valid for the time period you investigate here. In the end, you argue that the magnitude of the emissions is ok because the
- *agreement between model results and observations indicates that. This seems to be inconsistent.* **Author's response:** Indeed, the total PM10 mass emitted from the residential combustion section, provided by the TNO database, was not altered, as on a daily basis there was good agreement between model results and measurements of total PM10, especially during cold days/smog events. However, the carbonaceous PM
- 30 components (OA and BC) where poorly captured. Thus, our improvements involved a different chemical profile to the residential combustion aerosol emissions (different emitted mass of OA and BC component of PM10) than the TNO (central heating profile), driven by the respective, specialized (RWB) measurements (p. 6 line 27- page 7 line 2).
- 9. page 6, line 15: What is the reason for the differences between your Mie calculations and the values used in COSMO-ART? Is the conclusion that the optical properties for soot are incorrect in COSMO-ART?
   Author's response: No, the optical properties for aerosol particles incorporated in COSMO-ART are correct. Nevertheless, they have been defined for typical air pollution conditions in an area of Germany. Our calculations are based on local conditions with respect to the aerosol chemical profile for the event/area of interest, as well as
- 40 for the local, representative RH and PBL height (cf. Sect. 2.4). Thus, they differ from the ones in COSMO-ART, but they are similar both to other findings in the Mediterranean (Mishra et al., 2014; 2015), as well as to respective model inputs (Takemura et al., 2002). The relevant phrase is now enriched as follows: "These values differ from the ones used in Vogel et al. (2009;

black and red lines in Figure 3), which is expected due to the different geographical areas and periods of interest between the two studies." (p.7 lines 31-32).

10. page 6, line 20: "The period studied can be characterized as a relatively mild winter period.": Why did you choose this period when you can expect that the heating activities will be comparably low?

Author's response: Indeed, the winter of 2013-14 was mild. Nevertheless, the period we have focused our research on the RWB implications on air pollution issues (19 Dec - 5 Jan) can be characterised as an intense
5 smog period, as explained in p. 8 lines 18-23). We have rephrased the initial statement (p. 8 line 5), so that the reader is not confused with respect to the whole winter and the selected smog period.

11. page 8, line 24/25: Is the intense smog period the entire period from 19 Dec to 21 Jan? The mean concentration values for case 2 agree better to the observations than those from the baseline although the total emission amount was the same. Is that correct? You should make this clear.

- **Author's response:** No, the smog period is not the entire modelled period. Taken from the text (p. 8 lines 18-20): "19 December 2013 05 January 2014 … excluding the rainy days mentioned above, constitutes the intense smog period of the current modeling study". Yes, the reviewer is correct, a clarification and explanation is needed for this finding. In fact, although the total PM10 mass emitted by residential combustion is unchanged within a
- 15 typical week, the weekday mass amounts are reduced due to the revised weekly cycles, which are instead emitted during the weekends (cf. Fig. 1b). Given the fact that the intense smog period (13 days) includes only 4.5 weekend days, the differences found by the two scenarios in the average PM10 concentrations are expected. This is now explained in the revised text (p.9 lines 13-16).
- 20 12. page 9, line 12 and line 15: You should avoid descriptions like "satisfactorily represented" and "nicely captured". Each reader might think differently about what is satisfying or nice. Obviously, for case 2 the temporal profile of the concentrations fits better to the observations than for case 1 and you should just describe this. You could give a correlation coefficient to show this in numbers.
- Author's response: 'satisfactorily represented' is replaced by 'reproduced' and 'nicely' has been removed.
  25 Correlation coefficients are added for the daily cycles of carbonaceous species, and mean hourly bias is added in the peak comparisons (p. 10 lines 3-4, lines 6-8).

13. page 10, line 33: "The peaks observed at the urban core, as revealed from this model application, are somewhat displaced from the exact location of the site. This finding is in line with the characterization...": This is a bit over-interpreted, having the coarse emission map in mind. The grid cells with the highest emissions are

30 *is a bit over-interpreted simply north of Thissio.*

10

Author's response: We agree over possible over-interpretation, which is why we have calculated the mean, minimum and maximum values of a greater urban area (118 km2, 15 cells; SW corner:  $37.94^{\circ}$ ,  $23.67^{\circ}$ ) and compared them to the model results at the grid point of Thissio. The results, which are given in p. 11 line 30 - p.

35 12 line 5), further support the argument already posed by Gratsea et al. (2016), on the characterization of Thissio as an urban background site, not intensively affected by local traffic and representative of the average background pollution conditions in Athens.

page 12, line 9/10: "... for a pre-crisis period or significant change in dominant heating fuels, the TNO MACC\_II emissions from residential heating should be further adjusted.": Why for a pre-crisis period? Didn't you say that the fuels changed significantly during the crisis? Why is the TNO inventory then wrong in a pre-crisis period?

Author's response: It is true that during the pre-crisis period, Greek households did not use wood as the primary heating fuel, but central heating installations, which emit far less  $PM_{10}$  mass than wood combustion. As evident

45 from our study, TNO-MACC\_II PM10 emission rates for Greece, better fit to this RWB increase, rather than the central heating emission conditions. Thus, caution should be placed when the simulated periods reflect non-wood

residential combustion, but the usage of other fuels, such as gas. The reason behind these findings might be that for the case of PM10 and PM2.5 in Greece, TNO had no data available, thus applied model approximations for the calculation of emissions (cf. reply to Comment no 5).

- 5 15. page 14: I do not agree with the last part of the conclusions. In particular: "Thus, human health implications, as well as policy making, when the fireplaces are in use to cope with high heating demand conditions (HD > 7.5 C), can be satisfactorily estimated and planned with the aid of such a tool.": You did not estimate any human health effects with your model system. How will help to estimate them? What can be planned with the help of the model results? "For mild winter conditions (Tmin > 8-9 C), a post-processing of
- 10 model results according to the linear regression between HD and model bias, can further improve the quality of the model system.": Wouldn't it be better to improve the emission estimates than to correct the model? -"Alternatively, an interactive treatment of RWB emissions, i.e. their online adjustment according to the actual temperature conditions of the simulation period, is proposed as a means to further enhance the reliability of operational forecasts of online-coupled atmospheric models." I agree to this, but how would you know how much wood is used in comparison to oil or other fuels?

Author's response: The reviewer is correct; we are not estimating health effects within this study. Nevertheless, these or similar model results can be used to provide maps with exceedances of the EU limits of PM, thus to estimate the differential exposure of population to air pollution threats. Such information could be of help to policy makers as well, with respect to warnings and measures against RWB emissions and population exposure during wintertime.

20 during wintertime.

Yes, we agree that the best approximation would be to improve emission estimates (online) than correct the model outputs (offline). In other words, a hybrid model of RWB treatment of emissions during the model runs, could be beneficial, depending on the scope of the study. The proposed methodology does not need the information with respect to the use of wood in comparison to other fuels, because it will adjust the given TNO

- 25 mass emissions according to the ambient temperature (timestep or hourly model values), based on actual measurements (PM vs. HD) that already include this behavioural response to the use of the different heating fuels. Such a modification is out of the scope of the current study, but valuable in future studies, targeting long-term wintertime periods or operational forecasts rather than intense and specific smog episodes. A supporting statement to this, is now added in p. 12 line 30).
- 30

**Minor comments**

16. page 1, line 17: ": : : accurately predicts : : :": Qualifiers like "accurately" are always a bit difficult if do not give numbers for the deviations.

Author's response: 'accurately predicts' is replaced by 'reproduces the measured'

**35**

17. *page 2, line 2: "exorbitant" should be replaced by "very high".* Author's response: done

18. page 3, line 7: Denier van der Gon

40 Author's response: done

19. page 2, line 21 and in the following: You use BC and EC as it is the same. You need to explain this a bit, e.g. provide definitions of both of them and their relation to what you call "soot"

Author's response: The reviewer is of course correct about the different definitions of black carbon, elemental carbon and soot. Apart from this specific statement in the Introductory section, we only deal with BC data throughout the current study. Therefore, we see no actual need for the provision of the definitions and interrelation, rather than providing a relevant reference and a statement with respect to their erroneous usage as synonyms (p. 2 line 22).

20. *page 3, line 32: " : : : which was, however, nonlinear.": In which way nonlinear?*

5 Author's response: The ratio of  $\Delta F$  (net radiative flux in W/m-2) and AOD (at 450 nm), has a linear relation for the net shortwave flux at the surface, i.e. the points in a scatter plot (y-axis  $\Delta F$ , x-axis AOD) form a line with a negative slope (Stanelle et al., 2010). For the longwave the graph is not a line and resembles more a logarithmic graph.

The text is now written as: "Stanelle et. al. (2010) concentrated on dust episodes over West Africa and found an average increase of 70 Wm-2 for the long-wave radiative effect. Comparing the AOD (at 450nm) with the surface long-wave flux, he found the relationship to be nonlinear in contrast to the shortwave case," (p. 33 - p. 4 line 2)

21. page 8, line2: "mean maximum nighttime PM10": over which period was the mean taken? Was this hourly?

- 15 **Author's response:** the period is 19 December 2013 05 January 2014. Yes, this is an hourly value. The whole phrase now is "The meteorological conditions favored the accumulation of smog over Athens during the period 19 December 2013 05 January 2014 (mean maximum nighttime hourly  $PM_{10}$  measured concentrations of 103.7 µg m-3)." (p. 8 lines 18-19).
- 20 22. page 9, line 1: "Overall, the revised run improved more than the 70% of the day and nighttime PM10 peaks during the intense smog period.": This is not well formulated and therefore unclear. What exactly is improved?

**Author's response:** The reviewer is correct. The phrase is now formulated as: "Overall, the revised run shows improvements in the calculated PPEA values for more than the 70% of the day and nighttime PM10 peaks during the intense smog period." (p. 9 lines 25-26).

page 9, line 10/11: "which leads to the improvement of the half PMIOA and of all PMIBC the daytime peaks during the intense smog period." This sentence is obscure.

Author's response: it is again about better PPEA values. The phrase is formulated as "The PPEA values for both carbonaceous species are significantly lower for case 2, i.e. for the half PM1OA and of all PM1BC the daytime peaks during the intense smog period." (p. 9 line 33 – p. 10 line 2).

23. page 9, line 28-30: "Thus, the aerosol chemical composition during the economic crisis is completely altered with respect to the chemical profile of wintertime aerosols beforehand." What was the chemical profile before the economic crisis?

Author's response: This is given in p. 6 lines 27-28: "...the original chemical profile of aerosol emissions from non-industrial combustion (20% BC, 40% OA and 40% others) was modified..."

24. page 10, line 2/3: "The comparison with available measurements during increased wood burning in the
Alpine area (Szidat et al., 2009) reveals similarities": Where are these measurements shown? Are they given as
the observations in Table 3? Please add the values they found somewhere.
Author's response: The values are now added in Table 5.

25. page 10, line 27: "PMIBC is found below 8": The unit is missing.
45 Author's response: The unit "µg m-3" is now added.

25

- 26. page 12, line 2/3: "Inversely, during the cold days (Tmin below 8 C) that the HD is increased (>7.5 C), and consequently no model overestimation occurs." Omit the word "that". In addition: It is true, that no overestimation occurs. Instead, you see an underestimation.
- Author's response: "that" is deleted from the text. We kept the rest of the sentence as was, since our main point in this figure and paragraph is model overestimation, i.e. that the model does not overestimate PM concentrations during smog events (high HD conditions) as during the milder winter days, without actually implying that it systematically underestimates (model bias below -25% occurs in 4 cases).
- 27. Table 3 is really hard to read. It looks unstructured I would suggest to split into two or three different tables. One suggestion could be to have the ratios in the bottom (OC/BC,...) in separate table and also the peak analysis in another separate one. For the ratios in the bottom, it is unclear how they are given. OC/BC = 2.8makes some sense, but what about the other values (BC/TC,...)? Are they given in %? BC/TC = 28 makes no sense. In addition, it seems that you derived them from case2/case 3 differences but you do not say this in the caption or anywhere else in the table itself.
- 15 **Author's response:** The table is now split in 3, as suggested by the reviewer. All ratios are now dimensionless (pure numbers). RWB fraction predictions are the outcome of case 2 case 3 differences. This is now explained in Table 3.

**28. Figure 7: The caption is wrong. It contains parts that belong to Fig. 8.**

20 Author's response: The caption is now corrected: "The mean daily cycle of PM10 concentrations during the mild winter period, from the observations (black dots) and model predictions by case 3 (green line). All data refer to the Thissio site (Athens)."

**Comments from Anonymous Referee #2:** The manuscript concerns the impact of increasing burning for residential heating in Athens, Greece. Even if the Greek situation is peculiar due to the impact of the economic crisis, this subject has great importance over the whole Europe, where the use of wood burning for house heating

5 is increasing in a number of areas of different states. Even if the scientific knowledge of the dangers associated to biomass burning is growing, the air quality impact of biomass burning in domestic devices is not yet perceived and understood by the population and decision makers.

The manuscript is properly organized and well written. It points out that biomass burning for house heating has a major impact on the air quality and on atmospheric composition, while it has a minor influence on local radiative forcing. The authors describe their approach to improve wood burning description within the emission modelling that could be useful for model development and application in different areas and contexts.

29. Because the manuscript is proposed for publication in a special issue on coupled chemistry-meteorology modelling, a discussion on the relevance of the on-line coupling approach on the presented results concerning atmospheric composition and aerosol chemistry is presently missing. The on-line coupling advantage is evident only for the studied aerosols feedback on meteorology, while it should be specified is any difference could be noticed on pollutants concentrations when feedback was switched off (Scenario 4).

Author's response: We thank the reviewer for the positive comments and the interesting suggestion. We have calculated the differences on PM10 concentrations (and their chemical composition) between Case2 and Case4. The feedback of the online coupling approach on aerosol is found very small, which was expected given the small negative effect of particles on radiation in our case study.

The section 3.2 is now named after "Impact of RWB smog on radiation and feedbacks on atmospheric composition", and a relevant discussion is now added therein (p. 13 lines 21-26), plus some additions in abstract (p. 1 line 24) and conclusions (p. 14 lines 32-33).

The present form of the manuscript needs a revision including: clarifications, figures improvement, extension of feedback effects discussion.

Specific comments:

30

25

2.2 Model framework and setup

Line 3

30. The model domain is defined to be "the extended area of Greece" (the same definition is repeated in
Table 1). This definition is quite generic and should be made more specific adding a Figure or a better definition of the domain boundaries.

Author's response: the definition of the horizontal domain in degrees is now added in Table 1.

Lines 5-6

40 31. The sentence "The atmospheric pressure...." is not understandable in this form. How where pressure and precipitation optimized? Do the authors refer to the choice of the microphysics scheme? What is the mentioned optimization?

Author's response: In the COSMO-ART (and pre-processor) configuration, there are certain parameters/flags that are spatially-sensitive. Among others, these settings correspond to a better balancing of the pressure fields

45 and to a tuning in precipitation, in accordance to the horizontal spatial resolution of 0.025 deg. Therefore, the input data remained intact, the microphysics scheme was not altered, but e.g. the mask for smoothing of steep

orography is adjusted, the critical value for normalized over-saturation is adjusted and so on. It should be noted that these changes are imposed either by using 'true' or 'false' or by applying certain code numbering in specific flags of the model (and pre-processor) job scripts. In order to avoid further puzzling of readers, we have decided to remove the respective phrase from the document.

**5**

Line 9 32.

Does "constant initial conditions" mean uniform initial conditions?

Author's response: Combining this with the next comment (no. 33), the phrase (p. 5 line 14) is now replaced by: "...the uniform (in space) and constant (in time) initial conditions (e.g. for SO2 and aerosol species)."

**10**

Table 1

33. It is not clear how the initial and boundary conditions for aerosols are defined. Are the values included in Table 1 uniform in space? Are those values kept constant at boundaries?

Author's response: Yes to both. This is now better explained in text (cf. response to Comment no. 32).

**15**

2.3 Modifications of the aerosol emissions

*34. Figure 2 is hardly understandable. Its quality should be improved.*

Author's response: all figures will be uploaded separately in high quality, following the ACP guidelines.

**20 Page 6**

Lines 21-23

35. Does the sentence "Combined with the temporal..." refer to Figure 2?

**Author's response:** After this comment, the phrase (p. 7 lines 3-4) is modified for clarity as: "...in Figure 2. These rates, combined with the temporal profiles (Figure 1),..."

**25**

36. Does Figure 2 describe average emissions or do plotted values refer to a specific hour? Author's response: These are hourly emissions "for a night hour of a weekday (Tuesday, 21.00 UTC)", as indicated in the caption of the figure.

**30 Lines 23-24**

37. Maximum wood burning emissions are said to be located at the urban core, while it would be reasonable to expect to have maximum emissions over peripheral areas, where the access to wood should be easier that in the center.

Author's response: In principle, population density is used as a spatial proxy in order to distribute wood combustion emissions (TNO report, 2010). The TNO-MACC\_II emission database we have used for this study

(Kuenen et al., 2014), uses a population map at high resolution, and a special proxy for the distribution of residential wood combustion. The latter takes into account both the population density, but also the proximity to wood (for more information on proxies for RWB, cf. Comment no. 7). Despite this combination for the distribution of residential wood combustion, an overallocation of the emissions in urbanized centres is possible,

40 as the reviewer also comments, which is already described in previous studies (Denier van der Gon et al., 2014 and Timmermans et al., 2013). We have now included this info in the text. (p. 7 lines 5-10).

2.4 The aerosol optical properties

45 Page 7

Line 9

38. Concerning aerosols composition, it is not clear how the concentration values reported in brackets should be interpreted.

Author's response: These are RWB smog period-averages of the respective hourly values measured at Thissio during the RWB periods of winter 2013-14. This is explained in the text, in the previous sentence: "...is based on

- 5 observational data collected in Greece during the 2013-2014 RWB smog episodes. In particular, the average surface chemical composition of ultrafine aerosols in Athens (pure soot: 2.8 μg m-3, water soluble mixture of sulfate, nitrate, ammonium and organics: 22.2 μg m-3), local relative humidity (50-70 %) and an average mixing layer height (600 m a.g.l., Gerasopoulos et al., 2017), were used to feed the OPAC software (Hess et al., 1998), which then by applying the Mie theory provides the respective optical properties for 61 wavelengths between
- 10 0.25 and 40 µm". Thus, no interpretation should be extracted from these values; we provide information on the representative (average) values we used to calculate the aerosol optical properties for our case study.

**Lines 15-18**

*39.* The values used for Athens differ from those used by Vogel et al. (2009). Is the difference due to the geographic area of application or to any other understandable reason?

**Author's response:** Yes, the difference is related both to the geographic area and to the selected events. The phrase (p. 7 lines 31-32) is now enriched as follows: "These values differ from the ones used in Vogel et al. (2009; black and red lines in Figure 3), which is expected due to the different geographical areas and periods of interest between the two studies."

20

15

**40. To which geographic region do values reported by Takemura et al. refer?**

**Author's response:** They do not correspond to a specific area, but are used as model inputs for the model study of Takemura et al. (2002). Thus, they are directly comparable to our findings. This is now more clearly explained in p. 8 line 2).

25

3.1 Impacts of residential wood burning (RWB) on atmospheric aerosol mass and chemistry

3.1.1 Aerosol model performance under smog influence

Line 23

30 41. Do values reported in Table 3 as "daily mean" refer to the whole period mean as understood from the manuscript text?

Author's response: Yes, these are averaged daily mean values for the RWB smog sub-period of the simulation period. The phrase 'averaged daily mean' is now used instead of 'daily mean'.

**35 Page 9**

Lines 1-2

42. How (on the basis of what parameters) is it evaluated the mentioned 70% improvement?

Author's response: The relevant phrase is now written as "Overall, the revised run shows improvements in the calculated PPEA values for more than the 70% of the day and nighttime  $PM_{10}$  peaks during the intense smog period.", so that this question is answered in the manuscript (p. 9 lines 25-26).

**Lines 10-11**

43. The meaning of the sentence "which leads to the improvement of the half PM1 OA and of all PM1 BC the daytime peaks during the intense smog period" is not clear.

Author's response: The relevant phrase is now written as "The PPEA values for both carbonaceous species are significantly lower for case 2, i.e. for the half  $PM_1OA$  and of all  $PM_1BC$  the daytime peaks during the intense smog period", so that its meaning is clear (p. 9 line 33 – p. 10 line 2).

5 3.1.2 Representative spatial aerosol fields

Lines 16-17

44. The reference to PM10 EU alarm threshold is not clear, please include proper references to EU directives.

10 Author's response: The reviewer is correct. The PM10 alarm value we refer to, is not suggested by any EU directive, but rather by the National Legislation (Joint Ministerial Decision by FEK 3272B/23-12-2013, in Greek), which supplements the respective EU directive with respect to public information and emission reduction thresholds.

The acronym 'EU' is now replaced with 'National' within text and a proper reference is added (p. 11 line 9). The same changes are now done for the alert threshold for the whole population (p. 6 line 29).

3.2 Impact of RWB smog on radiation Page 12 Lines 25-27

20 45. The sentence concerning removal of absorbing BC is not very clear and should be rephrased to be more clearly understandable.

**Author's response:** This sentence (p. 13 lines 18-20) is now rephrased as: "...while the daytime mean was 98  $\mu$ 
[revised manuscript text omitted]

(%) |
|------------------|-----------------------------------------------------------|-------------------------------------|---------------------|-----------------------------|---------------------------------------------|---------------------|---------------------|----------------------------|---------------------------------------------|-----------------------|---------------------|
|                  |                                                           | Averaged daily mean                 | 45.2         | 58.7                 | 15.0                                 | 0.24         | 95           | 50.8                | 19.3                                 | 0.66           | 83           |
| I
(3   | PM10 μg m-3
332 samples)  | (smog period)
std. deviation     | 33.1         | 28.8                 |                                             |                     |                     |                            |                                             |                       |                     |
|                  |                                                           | RWB fraction (%)* |                     |                             |                                             |                     |                     | 51                  |                                             |                       |                     |
|                  |                                                           | Averaged daily mean                 | 28.4         | 18.8                 | -8.2                                 | 0.39         | 65           | 34.4                | 6.6                                  | 0.73           | 118          |
| P ]
(* | M1OA µg m-3
333 samples)  | (smog period)
std. deviation     | 30.9         | 8.6                  |                                             |                     |                     | 22.4                |                                             |                       |                     |
|                  |                                                           | RWB fraction (%)* |                     |                             |                                             |                     |                     | 78                  |                                             |                       |                     |
| P1
(2  |                                                           | Averaged daily mean                 | 4.6          | 9.9                  | 5.2                                  | 0.47         | 236          | 7.1                 | 2.5                                  | 0.53           | 121          |
|                  | PM1BC µg m-3
212 samples) | (smog period)                       |                     |                             |                                             |                     |                     |                            |                                             |                       |                     |
|                  |                                                           | std. deviation                      | 3.8          | 6.7                  |                                             |                     |                     | 4.7                 |                                             |                       |                     |
|                  |                                                           | RWB fraction (%)* | 45           |                             |                                             |                     |                     | 42                  |                                             |                       |                     |

\*RWB fraction predictions are the outcome of case 2 – case 3 differences. Measurements for the same fraction are only provided for BC.

| ¢                             | <del>omponent</del> | Parameter                         | Observations | <del>Case 1</del>
<del>(baseline)</del> | <del>Mean</del>
hourly
bias | F 2 | MANGE
(%)    | <del>Case 2</del>
( <del>revised)</del> | <del>Mean</del>
hourly
bias | F 2 | MANGE
(%)    |
|-------------------------------|---------------------|-----------------------------------|---------------------|--------------------------------------------|-----------------------------------|-----------------------|-----------------|--------------------------------------------|-----------------------------------|-----------------------|-----------------|
|                               | 4                   | daily mean (smog period)          | 4 <del>5.2</del>    | <del>58.7</del>                            | <del>15.0</del>                   | <del>0.2</del> 4      | <del>95</del>   | <del>50.8</del>                            | <del>19.3</del>                   | <del>0.66</del>       | <del>83</del>   |
|                               | $\frac{1}{1}$       | std. deviation                    | <del>33.1</del>     | <del>28.8</del>                            |                                   |                       | PPEA (%) | <del>32.7</del>                            |                                   |                       | PPEA (%) |
| PM                            | ⊨ ™ ¶ ∰  | <del>mean daytime maximum</del>   | <del>47.8</del>     | <del>115.5</del>                           |                                   |                       | <del>173</del>  | <del>78.1</del>                            |                                   |                       | <del>80</del>   |
|                               | 40 s                | <del>mean nighttime maximum</del> | <del>103.6</del>    | <del>80.9</del>                            |                                   |                       | -8              | <del>110.1</del>                           |                                   |                       | <del>23</del>   |
|                               | %                   | RWB fraction               |                     |                                            |                                   |                       |                 | <del>51</del>                              |                                   |                       |                 |
|                               |                     | daily mean (smog period)          | <del>28.4</del>     | <del>18.8</del>                            | <del>-8.2</del>                   | <del>0.39</del>       | <del>65</del>   | <del>34.4</del>                            | <del>6.6</del>                    | <del>0.73</del>       | <del>118</del>  |
| 24                            |                     | std. deviation                    | <del>30.9</del>     | <del>8.6</del>                             |                                   |                       | PPEA (%) | <del>22.4</del>                            |                                   |                       | PPEA (%) |
| * <del>V0*Md</del> |                     | <del>mean daytime maximum</del>   | <del>23.5</del>     | <del>23.9</del>                            |                                   |                       | <del>119</del>  | <del>39.3</del>                            |                                   |                       | <del>103</del>  |
|                               | 99 s                | <del>mean nighttime maximum</del> | <del>85.1</del>     | <del>19.6</del>                            |                                   |                       | <del>-58</del>  | <del>69.1</del>                            |                                   |                       | <del>9</del>    |
|                               | <mark>₩</mark>      | RWB fraction                      |                     |                                            |                                   |                       |                 | <del>78</del>                              |                                   |                       |                 |

| d                   | daily mean (smog period)          | <del>4.6</del> | <del>9.9</del>  | <del>5.2</del> | <del>0.47</del> | 236            | 7.1             | <del>2.5</del> | 0.53 | 121             |
|---------------------|-----------------------------------|----------------|-----------------|----------------|-----------------|----------------|-----------------|----------------|------|-----------------|
|                     | std. deviation                    | <del>3.8</del> | <del>6.7</del>  |                |                 | PPEA (%)       | <del>4.7</del>  |                |      | PPEA (%) |
|                     | <del>mean daytime maximum</del>   | <del>1</del> 4 | <del>27</del>   |                |                 | <del>590</del> | <del>15</del>   |                | -    | <del>258</del>  |
|                     | <del>mean nighttime maximum</del> | <del>5.2</del> | <del>12.6</del> |                |                 | <del>63</del>  | <del>11.3</del> |                |      | <del>38</del>   |
| %                   | RWB fraction               | 4 <del>5</del> |                 |                |                 |                | <del>42</del>   |                |      |                 |
| OC/BC               | daily mean (smog period)          | <del>2.9</del> | <del>1.1</del>  |                |                 |                | <del>2.8</del>  |                |      |                 |
| BC/TC               | daily mean (smog period)          | <del>28</del>  | 47              |                |                 |                | <del>27</del>   |                |      |                 |
| TC/PM 10 | daily mean (smog period)          | <del>62</del>  | <del>48</del>   |                |                 |                | <del>61</del>   |                |      |                 |
| OA/PM 10 | daily mean (smog period)          | <del>50</del>  | <del>32</del>   |                |                 |                | <del>51</del>   |                |      |                 |
| BC/PM 10 | daily mean (smog period)          | <del>11</del>  | <del>17</del>   |                |                 |                | <del>11</del>   |                |      |                 |

\* Organic aerosol (OA) predictions are divided by 1.6 (Turpin and Lim, 2001), to extract the carbon mass (OC) used for the calculation of the OC/BC ratio.

Table 4: Mean daytime/night-time maximum values and prediction skill metrics of the aerosol concentrations against ground measurements (at Thissio) during the RWB smog period (19 December, 2013 – 05 January, 2014) in Athens. Numbers in bold represent the calculated statistics, given by equation A3.

| Component       | Time frame | Observations | Case 1
(baseline) | PPEA
(%) | Case 2
(revised) | PPEA
(%) |
|------------------------|-------------------|---------------------|-----------------------------|--------------------|----------------------------|--------------------|
| $PM_{10} \mu g m^{-3}$ | daytime    | 47.8         | 115.5                | 173         | 78.1                | 80          |
| (332 samples)          | night-time        | 103.6        | 80.9                 | -8          | 110.1               | 23          |
| $PM_1OA \mu g m^{-3}$  | daytime    | 23.5         | 23.9                 | 119         | 39.3                | 103         |
| (333 samples)          | night-time | 85.1         | 19.6                 | -58         | 69.1                | 9           |
| $PM_1BC \mu g m^{-3}$  | daytime    | 14           | 27                   | 590         | 15                  | 258         |
| (212 samples)          | night-time | 5.2          | 12.6                 | 63          | 11.3                | 38          |

Table 5: Averaged daily means of selected aerosol mass fractions against ground measurements at Thissio (Athens) during the RWB smog period (19 December, 2013 – 05 January, 2014) and at Swiss Alpine areas (winter 2005; Szidat et al., 2009). Organic aerosol (OA) predictions are divided by 1.6 (Turpin and Lim, 2001), to extract the carbon mass (OC) used for the calculation of the OC/BC ratio.

10

| iviaea by | 1.0 ( | I urpin and | Lim, 2001), | to extract | the carbon mas | s (OC) used I | or the calculation | on of the UC/I | BC rai |
|-----------|-------|-------------|--------------------|------------|----------------|---------------|--------------------|----------------|---------------|
|           |       |             |                    |            |                |               |                    |                |               |
|           |       |             |                    |            |                |               |                    |                |               |
|           |       |             |                    |            |                |               |                    |                |               |

| Component          | Observations
Athens / Alpine area * | Case 1
(baseline) | Case 2
(revised) |
|---------------------------|---------------------------------------------------|-----------------------------|----------------------------|
| OC/BC                     | 2.9 / 3.7                                  | 1.1                  | 2.8                 |
| BC/TC                     | 0.28 / 0.23                                       | 0.47                 | 0.27                |
| TC/PM10 | 0.62 / 0.89                                | 0.48                 | 0.61                |

| OA/PM 10 | 0.50 / -    | 0.32        | 0.51 |
|----------------------------|--------------------|-------------|-------------|
| BC/PM10  | 0.11 / 0.10 | 0.17 | 0.11 |

---

## Author Response (AR2)

Below, the final author comments to the revised article on behalf of all co-authors are published. The response to Referee #2 is structured as: (1) comments from Referee (in italics), (2) author's response after each comment (normal fonts), (3) author's changes in manuscript, marked with track changes.

**Comments from Anonymous Referee #2:**
The manuscript has been improved and almost all the parts that needed a revision have been modified/integrated following the reviewers' comments and answering their questions.

The inclusion of a discussion concerning the feedback on atmospheric composition completed the work considering the general focus on coupled atmospheric modelling.

The manuscript can be published after the minor revision required to clarify few items highlighted by the following detailed comments.

Detailed comments

The Authors should consider to include in their discussion (either in section 2.3 or in the conclusions section) the possible improvement that could be obtained from an high resolution local emission inventory. Previous studies (e.g. Denier van der Gon et al., 2011) showed that the use of continental scale emission inventories for city scale modelling can be quite crude for large European cities.

Denier van der Gon et al., 2011, Discrepancies between top-down and bottom-up emission inventories of megacities: the causes and relevance for modeling concentrations and exposure, Air Pollution Modeling and its Application XXI, 199-204, Springer, Dordrecht

Answer: This is now discussed in the conclusions.

2.2 Model framework and setup

Line 13
Does "uniform (in space) and constant (in time) initial conditions" mean "uniform (in space) initial and boundary conditions and constant (in time) boundary conditions"?
The detail provided in Table 1 for initial and boundary conditions does not help much to understand the chosen approach. Which is the origin of the values indicated in Table 1? Minimum values, winter typical values? Do they come from measurements? Values in Table 1 are for the surface layer. Do you assume any form of vertical profile? Are the mentioned aerosol species the only ones considered for pm2.5 and pm10 IC and BC?

Answer: No, the start-up time is used to dampen the effect of initial conditions, therefore the part "constant in time" is now removed from text.
The initial and boundary aerosol values are typical background conditions used for all previous COSMO-ART applications from the first author, cf. Athanasopoulou et al., 2013;2014 and initially based on Fountoukis et al., 2011. Their vertical profile follows air density. Yes, IC and BC are only imposed on these secondary aerosol species, and specifically in their aged modes.
All the above information is now added to Table 1.

2.3 Modifications of the aerosol emissions

Figure 2 is hardly understandable. The color shaded areas are not well defined. A "pixel-style" plot would be more readable. The isopleths and their labels are substantially unreadable. It is not a matter of picture quality or graphic resolution, the way the picture is built can definitely be improved.

Answer: The picture is now re-built, using a pixel-style plot, with the isopleths and labels in a readable format.

Lines 8-9
Does the sentence "due to the switch from residential heating from fossil fuels to wood burning" stay for "due to the switch of residential heating from fossil fuels to wood burning"?

Answer: Yes, which is now corrected in text.

3.1.1 Aerosol model performance under smog influence

Line 22
Wrong open bracket.
Answer: the bracket is now deleted

Lines 28-29
I do not understand the meaning of the sentence "i.e. for the half PM1OA and for all PM1BC the daytime peaks occurred during the intense smog period.". What does "the half of PM1 OA and of all PM1 BC" mean?
Answer: We are referring to the half of the number of $PM_1OA$ daytime peaks, which is now corrected in the text.

3.2 Impact of RWB smog on radiation

Lines 21-23
The Authors say: "This slight increase in the concentration of aerosol species due to the aerosol-radiation interaction, is associated with an almost 10 m lower mean PBL height in case 4 compared to case 2, in accordance to the respective findings in Forkel et al. (2012)."
If case 4 has lower PBL height than case 2, I would expect to find higher concentrations for case 4 that for case 2, having to deal with near surface emissions. While you said that (case 2 – case 4) give positive results. What is the effective PBL height effect?
Answer: Yes, the reviewer is correct. This was a typo. In fact, the lower PBL height is found for case 2 when compared to case 4, which is now corrected in text.

[revised manuscript text omitted]